# Extracellular matrix in multicellular aggregates acts as a pressure sensor controlling cell proliferation and motility

Monika E Dolega[1], Sylvain Monnier[2], Benjamin Brunel[1], Jean-François Joanny[3], Pierre Recho[1]*, Giovanni Cappello[1]*

[1]Université Grenoble Alpes, Laboratoire Interdisciplinaire de Physique, CNRS, Grenoble, France; [2]Université de Lyon, Université Claude Bernard Lyon 1, CNRS, Institut Lumière Matière, VILLEURBANNE, France; [3]Collège de France, PSL Research University, Paris, France

**Abstract** Imposed deformations play an important role in morphogenesis and tissue homeostasis, both in normal and pathological conditions. To perceive mechanical perturbations of different types and magnitudes, tissues need appropriate detectors, with a compliance that matches the perturbation amplitude. By comparing results of selective osmotic compressions of CT26 mouse cells within multicellular aggregates and global aggregate compressions, we show that global compressions have a strong impact on the aggregates growth and internal cell motility, while selective compressions of same magnitude have almost no effect. Both compressions alter the volume of individual cells in the same way over a shor-timescale, but, by draining the water out of the extracellular matrix, the global one imposes a residual compressive mechanical stress on the cells over a long-timescale, while the selective one does not. We conclude that the extracellular matrix is as a sensor that mechanically regulates cell proliferation and migration in a 3D environment.

*For correspondence:
pierre.recho@univ-grenoble-alpes.fr (PR);
giovanni.cappello@univ-grenoble-alpes.fr (GC)

**Competing interests:** The authors declare that no competing interests exist.

## Introduction

Aside from biochemical signaling, cellular function and fate also depend on the mechanical state of the surrounding extracellular matrix (ECM) (*Humphrey et al., 2014*). The ECM is a non-cellular component of tissues providing a scaffold for cellular adhesion and triggering numerous mechanotransduction pathways, involved in morphogenesis and homeostasis (*Vogel, 2018*). An increasing number of studies in vivo and in vitro shows that changing the mechanical properties of the ECM by re-implanting tissues or changing the stiffness of the adherent substrate is sufficient to reverse aging (*Segel et al., 2019*), accelerate developmental processes (*Barriga et al., 2018*) or modulate tumor malignancy (*Paszek et al., 2005*; *Tanner et al., 2012*).

The importance of the mechanical context in cancer has been highlighted for a long time by experiments altering the composition and stiffness of the ECM (*Levental et al., 2009*). It has also been shown that the tumor growth is modulated by the mechanical compression caused by the tumor itself, as it expands in a confined environment (*Fernandez-Sanchez et al., 2010*; *Nia et al., 2017*). Such patho-physiological growth under pressure has also been studied in vitro. When multicellular aggregates are confined by soft gels (*Helmlinger et al., 1997*; *Alessandri et al., 2013*; *Taubenberger et al., 2019*) or submitted to a gentle osmotic compression (*Montel et al., 2011*; *Dolega et al., 2017*), their growth is substantially reduced. It has been demonstrated that the cell cytoskeleton is involved in the response to compression and can trigger the growth impediment through a cell-cycle inhibition (*Taubenberger et al., 2019*; *Delarue et al., 2014*). In addition, the cellular volume has been recently proposed to be a key parameter in the mechanosensitive pathway

(*Delarue et al., 2014*; *Han et al., 2020*). Nevertheless, it is not known how such mild global compression is transduced to the individual cells of the aggregate to alter their proliferation.

Here, we posit that cells mainly respond to the mechanical stress transmitted by the ECM, when the aggregate is under compression. This hypothesis is motivated by two evidences. First, an aggregate is a composite material made of cells, extracellular matrix and interstitial fluid. The presence of hydrated extracellular matrix is evidenced by the abundance of fibronectin in the interstitial space (*Figure 1a* and Appendix 8). As the ECM is 100- to 1000-fold more compressible than the cells, it absorbs most of the deformation, but still transmits the mechanical stress to the cells. Second, whereas an osmotic pressure of a few kPa strongly reduces the cell proliferation within multicellular aggregates, an identical pressure has no effect on individual cells cultured on a Petri dish, in the absence of ECM (*Montel et al., 2011*). In addition, the use of drugs affecting the cytoskeleton organization has a negligible effect on the effective compressibility of multicellular aggregates (Appendix 12). This indicates that the volume loss under compression is mainly due to ECM dehydration (*Dolega et al., 2021*).

To test the hypothesis that cells respond to the ECM deformation, we introduce an experimental method that uncouples the cell volume change from the mechanical stress transmitted to the cells through the ECM. We apply this method for both multicellular aggregates and individual cells embedded in a gelified ECM. In parallel, we present a theoretical framework to estimate both the displacement and the stress at the ECM/cell interface in response to an osmotic compression, and verify experimentally its qualitative prediction. At a longer timescale, we probe the effect of the ECM compression on the cellular response. In particular, we demonstrate that, even in the absence of cell deformation, the ECM alone regulates cell proliferation and motility.

## Results

### Selective-compression method

We developed a simple method to either selectively compress cells embedded in ECM or the whole aggregate composed of ECM and cells. This method is based on the use of osmolytes of different sizes. When big enough, the osmolytes do not infiltrate the ECM and thus compress the whole aggregate by dehydrating the ECM, which in turn mechanically compresses the cells (*Monnier et al., 2016*). When smaller than the exclusion size of the ECM, the osmolytes percolate through the ECM meshwork and compress the cells which can then pull on the ECM (see schematic in *Figure 1b–d* and Appendix 7). We already proved (*Montel et al., 2012*) that a gentle osmotic pressure $\Pi_d$ exerted using large dextran molecules considerably reduces the proliferation of cells inside multicellular spheroids. The effect was visible starting from $\Pi_d = 500\mathrm{Pa}$ and saturated at $\Pi_d \simeq 5\mathrm{kPa}$. Unless explicitly stated, the experiments described in this article were performed at $\Pi_d \simeq 5\mathrm{kPa}$, a value that minimizes the pressure, but exacerbates the biological effects.

We validated our approach by compressing ECM, cells and multicellular spheroids (MCS) using osmolytes with gyration radii $R_g$ respectively larger and smaller that the ECM pore sizes (*Figure 2*). As osmolytes, we used dextran molecules ranging from 10 to 2000 kDa. As a proxy of ECM, we used Matrigel (MG), a commercially available matrix secreted by cancer cells (*Kleinman and Martin, 2005*). To visualize the effect of the compression on the ECM, we prepared microbeads composed of matrigel, with a diameter of 100 µm (*Figure 2a* ). As shown in *Figure 2a* (top panel), fluorescent dextran molecules with a gyration radius below 5 nm (MW <70 kDa, hereafter called 'Small'; *Granath, 1958*) equally color the MG beads and the surrounding solution (left). Conversely, dextran molecules larger than 15 nm (MW >500 kDa, Big') do not penetrate inside the MG beads, which appear darker than the surroundings (right). By following the evolution of the bead diameter subjected to $\Pi_d = 5\mathrm{kPa}$ (measurements taken before the compression and 45 min after the compression), we observed that small dextran molecules compress the matrigel beads by 2.5 ± 0.7% of their initial volume (*Figure 2a*, middle and bottom panels). Conversely, the same pressure caused by big dextran molecules occasions a much larger compression of 63 ± 5% (*Figure 2b*). The relatively minor compression occasioned by small dextran can be explained by thermodynamic theories involving chemical interaction between the matrix and the permeating polymer (*Brochard, 1981*; *Bastide et al., 1981*), an aspect that we neglect in this article.

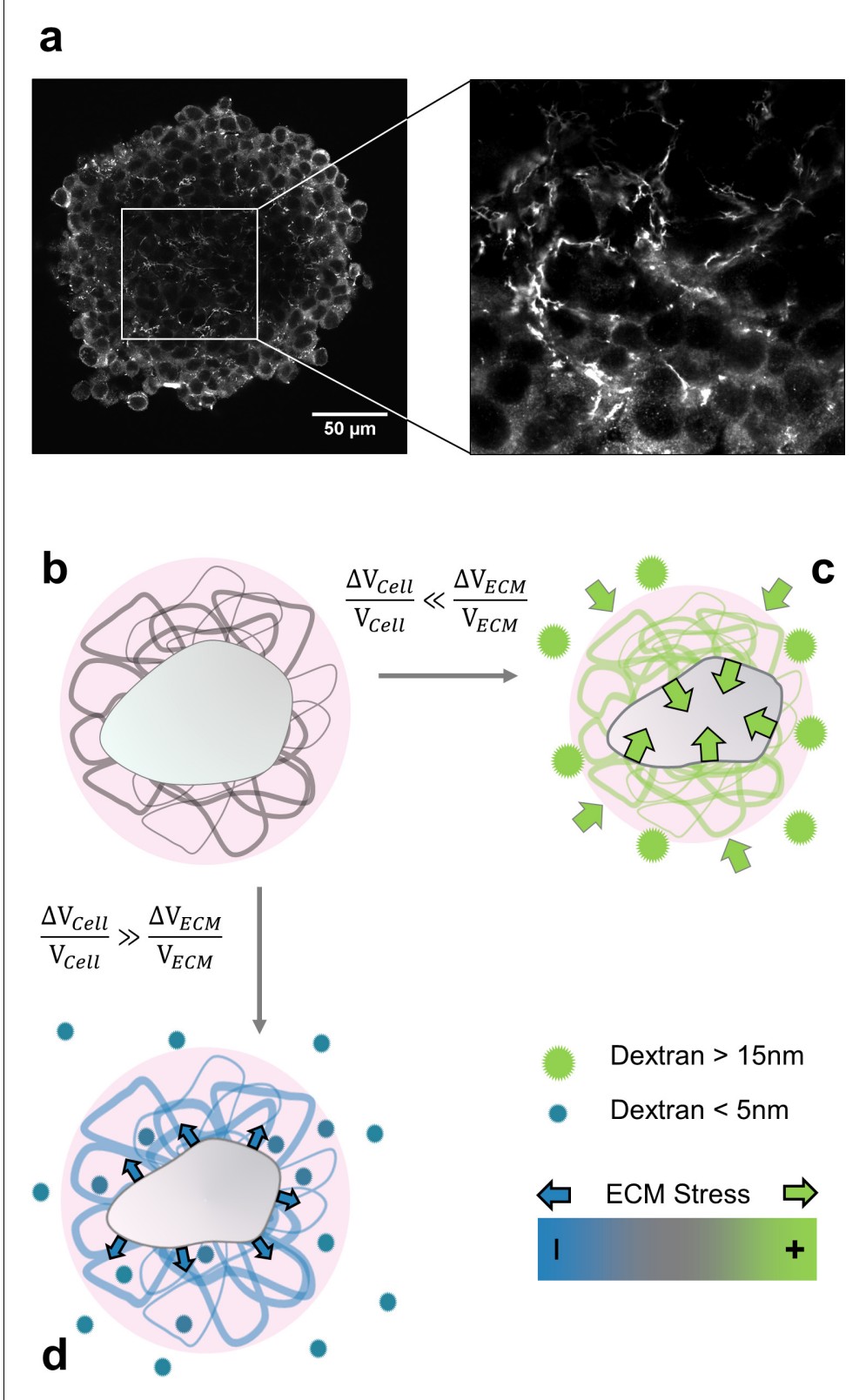

**Figure 1.** Selective compression method. (a) Immunofluorescent staining of fibronectin in the interstitial space of a multicellular spheroid made of CT26 cells (b) Schematic view of a cell (gray) embedded in extracellular matrix (filaments), permeated by interstitial fluid (light pink). (c) Big osmolytes (green) do not penetrate through the ECM and induce a global compression. Being much more compressible than the cells, the extracellular matrix absorbs

*Figure 1 continued on next page*

*Figure 1 continued*

most of the deformation and exert a positive stress on the cell. (d) Small osmolytes (blue) enter the ECM without exerting any osmotic pressure on it. Conversely, they compress the cell which, in turn, exerts a tension on the ECM.

Analogous experiments were performed using individual CT26 cells (murine colon carcinoma cells) and multicellular spheroids made with the same cell line. As the volume loss of individual cells is not measurable at $\Pi_d = 5$ kPa (*Monnier et al., 2016*), individual CT26 cells are submitted to $\Pi_d = 15$ kPa. At this pressure, we measured a relative compression $\Delta V_c / V_c = 3.8 \pm 0.8\%$ (*Figure 2c*), where $V_c$ is the cell volume and $\Delta V_c$ the volume loss upon the application of $\Pi_d$. This compression indicates that CT26 cells have an effective osmotic modulus $K_c = 400 \pm 100$ kPa. In contrast to single cells, MCS are much more compressible, as they lose up to 15% of their volume under an omostic pressure with big dextran of $\Pi_d = 5$ kPa (*Figure 2d*; See also *Dolega et al., 2021* for a detailed mechanical analysis). Furthermore, these measurements indicate that MCS have a typical effective osmotic modulus of $K_s \simeq 30$ kPa, 15-folds smaller than that of individual cells (*Dolega et al., 2021*). In contrast, small dextran molecules have no measurable effect on the volume of MCS, for moderate osmotic pressures (up to $\Pi_d = 10$ kPa). However, larger pressures with these small osmolytes can lead to a cell compression within the MCS associated with a swelling of the interstitial space as we show in Section 'Selective compression of ECM in multicellular spheroids'.

These results confirm the ability of our method to discriminate between the effects occasioned by the compression of the whole MCS, and those due to the compression of the cells alone within the aggregate.

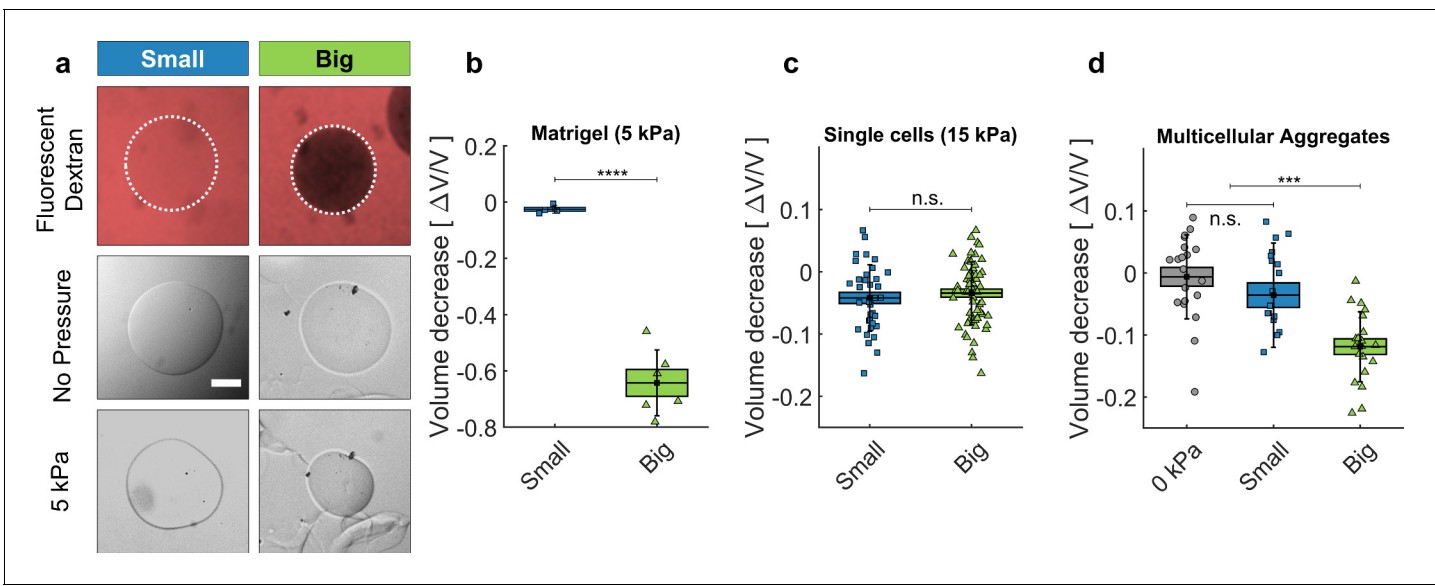

**Figure 2.** Cell and matrigel compression. (a) Fluorescently labeled dextran molecules only permeate the beads (top-left panel) if their gyration radius is smaller than 5 nm. Otherwise (top-right panel) they are larger than the exclusion size of the matrigel network and are excluded from the bead. Compression of MG beads, occasioned by dextran molecules of two different sizes (Small: 70 kDa; Big: 500 kDa). Phase contrast images taken before and after the addition of pressure. (b) Beads lose $63 \pm 5\%$ of their initial volume when compressed using big dextran, and $2.5 \pm 0.7\%$ with small Dextran. N = 10. (c) Compression of individual cells using dextran of different sizes, with $\Pi_d = 15$ kPa. At $\Pi_d = 5$ kPa the compressibility of individual cells is not measurable. Cell compressibility is thus negligible in comparison to that of Matrigel. (d) MCS compression under $\Pi_d = 5$ kPa, exerted using small (blue) and big (green) dextran molecules. In control experiments (0 kPa), the culture medium is replaced by fresh medium without dextran. (box : ±SEOM; error bars: ± SD • : single realizations).

The online version of this article includes the following source data for figure 2:

**Source data 1.** Data for *Figure 2b,c,d*.

## Theory: the effect of a selective compression applied to a cell nested in extracellular matrix

For simplicity, we consider the case of a single cell nested in a large -compared to the cell size- ball of ECM and subjected to the osmotic pressure $\Pi_d$ obtained by supplementing the culture medium with either small or big dextran. We assume that the small dextran can freely permeate in the ECM meshwork while the big one is excluded. Our aim is to compute the displacement of the cell boundary as well as the stress applied on the cell upon application of $\Pi_d$ in both conditions. Our model, detailed in Appendix 6.5, essentially couples a classical active pump-and-leak model (*Hoppensteadt and Peskin, 2012*) for the cell volume regulation through ion pumping and the constitutive behaviour of the ECM, which is assumed to be poro-elastic at a short timescale where remodeling is negligible. The cell cortex mechanics plays a negligible role in setting the cell volume since it involves stresses that are small compared to the osmotic forces. For simplicity, we neglect the mechano-sensitive nature of ion channels.

We show in Appendix A.5 that, for realistic estimates of the model parameters, the application of $\Pi_d$ with both small or big dextran leads to the same cell volume loss which does not involve the mechanical properties of the ECM but only the cell volume regulation system:

$$\frac{\Delta V_c}{V_c} = \frac{\Pi_d}{(1-\beta)\Pi_e}, \tag{1}$$

where $\Pi_e$ is the osmotic pressure of ions in the culture medium and $\beta \simeq 0.1$ is a non-dimensional parameter representing the active pumping of ions (see Appendix A.4). The relation (*Equation 1*) shows that the reduction of the cell volume under compression is mainly resisted by the active osmotic equilibration of ions through the cell membrane. For relatively low pressures ($\Pi_d \ll \Pi_e \simeq 500$ kPa), the relative change of volume $\Delta V_c/V_c$ is negligible. More quantitatively, formula (*Equation 1*) provides the estimate of the osmotic modulus of a cell $K_c = (1-\beta)\Pi_e \simeq 450$ kPa which is in agreement with the value measured for CT26 cells.

However, the mechanical stress applied by the ECM to the cell is qualitatively and quantitatively different in the two situations. For big dextran, this stress is compressive as the dominating effect of the dextran is to compress the ECM which in turn compresses the cells. Within some realistic approximations the amount of this compressive stress (the traction force applied by the matrix on the cell) can be approximated as the applied osmotic pressure:

$$T_{\text{big}} = -\Pi_d < 0. \tag{2}$$

In sharp contrast with the previous situation, for small dextran, the stress applied by the ECM on the cell is tensile. In fact, the dominating effect is that small dextran compresses the cells but not the ECM. Thus, cell compression is balanced by a tensile force in the ECM. This tension is given by

$$T_{\text{small}} = \frac{G \sim \Pi_d}{3(1-\beta)\Pi_e} > 0, \tag{3}$$

where $G$ is the ECM shear modulus. Formulas (*Equations 1, 2 and 3*) hold in the ideal case, where osmolytes do not interact with the matrix and the axisymmetric system has stress free boundaries at infinity (the ECM ball radius is much larger than the cell radius).

In practice, for a moderate osmotic shock $\Pi_d \simeq 5$ kPa, the dextran concentration is much smaller than the characteristic ion concentration of the external medium (few hundreds millimolars) and the tension can be considered negligible: $T_{\text{small}} \simeq 20\text{Pa} \ll T_{\text{big}} \simeq 5\text{kPa}$ because the ECM is soft. Therefore, in this condition, the presence of ECM makes the cell mechanically sensitive to a moderate osmotic compression using big dextran molecules, but not when using small dextran molecules. In both cases the cell volume is affected in the same negligible way, but the mechanical stress applied by small dextran on the cell is negligible compared to that exerted by big dextran.

If the osmotic pressure is further increased, the compression with small or big dextran can induce a measurable effect on the cell volume. However, the mechanical stress applied by the ECM to the cell remains fundamentally different in both situations: tensile for the small dextran and compressive for the big one.

## Selective compression of ECM in multicellular spheroids

To test our theoretical predictions that the interstitial space is compressed under dextran pressure, we injected individual MCS (4–5 days old) into a 2D confiner microsystem and let them relax for few hours (*Figure 3a*). The MCS were thus immobilized and partially flattened inside the 2D confiner. In order to follow the evolution of the interstitial space under an osmotic compression, the culture medium was supplemented with a fluorescent tracer. The interstitial fluorescence was measured using two-photon microscopy (*Figure 3b*). The images of the confined multicellular aggregates were normalized to the fluorescence of the external medium and segmented with a thresholding procedure, and the signal exceeding the threshold value was integrated over the whole aggregate to quantify the total fluorescence of the interstitial space (*Figure 3c*). Due to optical limitations, we emphasized the effect by increasing the applied osmotic pressure to $\Pi_d = 40$ kPa for small dextran and to $\Pi_d = 15$ kPa for the big ones.

In accordance with our theoretical predictions, we obtained two opposite behaviors, depending on the dextran size. Small dextran molecules induced a ~35 ± 10% increase in the fluorescence intensity in the interstitial space (*Figure 2c*) while the total volume of the aggregate was reduced by ~10% (*Figure 2d*). Simultaneously, the cell volume decreased (Appendix 9), thus stretching the ECM into occupying more interstitial space. In contrast, for big dextran we measured a loss of half the fluorescence, meaning that a large amount of interstitial liquid had left the intercellular space of the aggregate. The extracellular matrix was thus compressed as predicted by *Equation (2)* and the overall MCS volume of the whole aggregate was reduced by ~17% (*Figure 2d*). We argued in *Dolega et al., 2021* that the total volume reduction of the aggregate obtained with big dextran could be due mostly to the compressibility of the ECM, while the cells are quasi-incompressible. The volume reduction of the aggregate induced by a 15 kPa pressure did not differ much from the one obtained with a $\Pi_d = 5$ kPa compression, as the ECM was already fully squeezed at 5 kPa.

These results are consistent with our theoretical prediction that big and small dextran have an opposite effect on the matrix. The first puts the ECM under compression, while the latter puts the ECM under tension. Remarkably, in both cases, the cells within the aggregate undergo almost the same deformation.

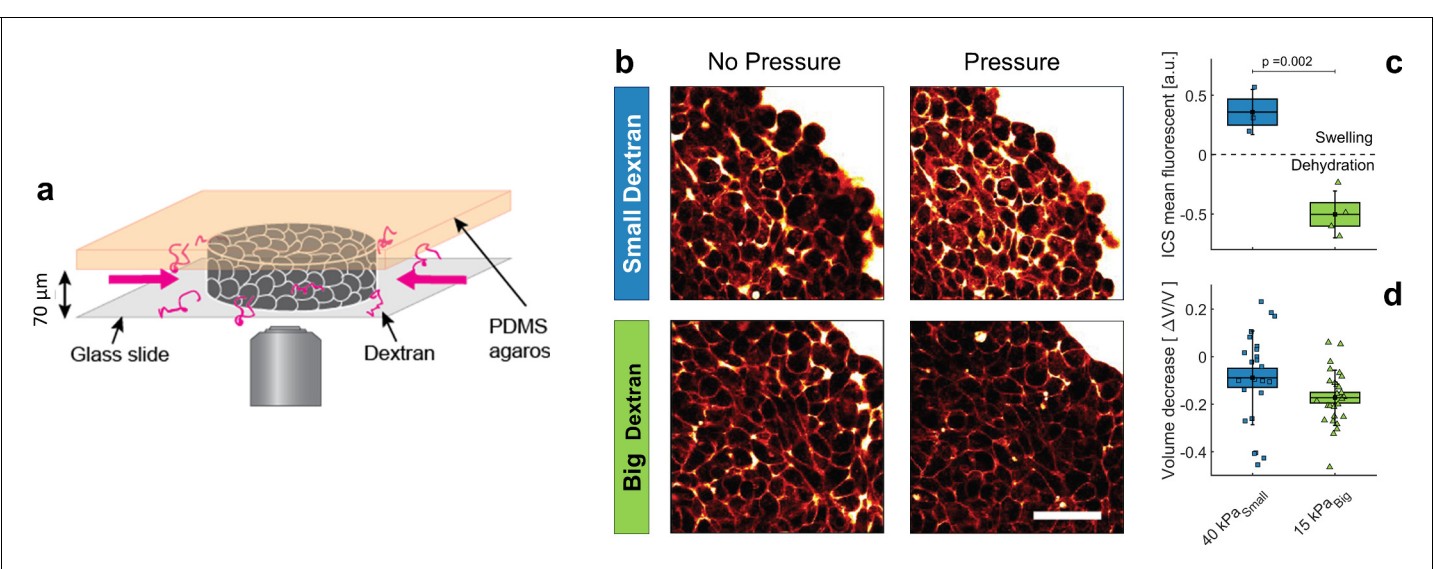

**Figure 3.** Effect of small versus big dextran on tissue intercellular space. (**a**) Schematic of the 2D confiner micro-device. The tissue is confined between the glass coverslip and the PDMS and does not move during medium exchange. (**b**) Two-photon images of the tissue before and after (20 min) osmotic shocks for dextran chains of 6 kDa (small) and 2MDa (big), for a given mass concentration of 100 g/L. Images were taken in the equatorial plane of the tissue, meaning 35 μm above the glass slide. Scale Bar: 50 μm (**c**) Mean fluorescence of the intercellular space averaged over the whole aggregate shown in panel b. (**d**) volume loss of spheroids submitted to $\Pi_d^{Small} = 40$ kPa (small dextran) and $\Pi_d^{Big} = 15$ kPa (big dextran).

The online version of this article includes the following source data for figure 3:

**Source data 1.** Data for *Figure 3c,d*.

## ECM compression controls cell proliferation and motility

To understand the role of the ECM on the cell fate at longer timescale, we assessed the proliferation and the motility of cells within MCS cultured in the presence of small and big dextran. *Figure 4a* represents the equatorial cryosections of spheroids in the three mechanical states ($\Pi_d$ = 0 kPa, $\Pi_d$ = 5 kPa small dextran, and $\Pi_d$ = 5 kPa big dextran). Proliferating cells were immuno-stained for Ki-67, a nuclear antigen present during the cell cycle, but absent in G0 phase (*Gerdes et al., 1984*). Whereas cells in control MCS ($\Pi_d$ = 0 Pa) present a rather uniform proliferation pattern, a global compression of MCS (big Dextran) stops cell division in the core and alters the overall MCS growth, as previously reported (*Helmlinger et al., 1997*; *Alessandri et al., 2013*; *Montel et al., 2011*). The density of Ki67-positive cells is reported in panel *Figure 4b*, as a function of the distance from the spheroid center and for the three conditions represented in panel *Figure 4a*. Remarkably, under

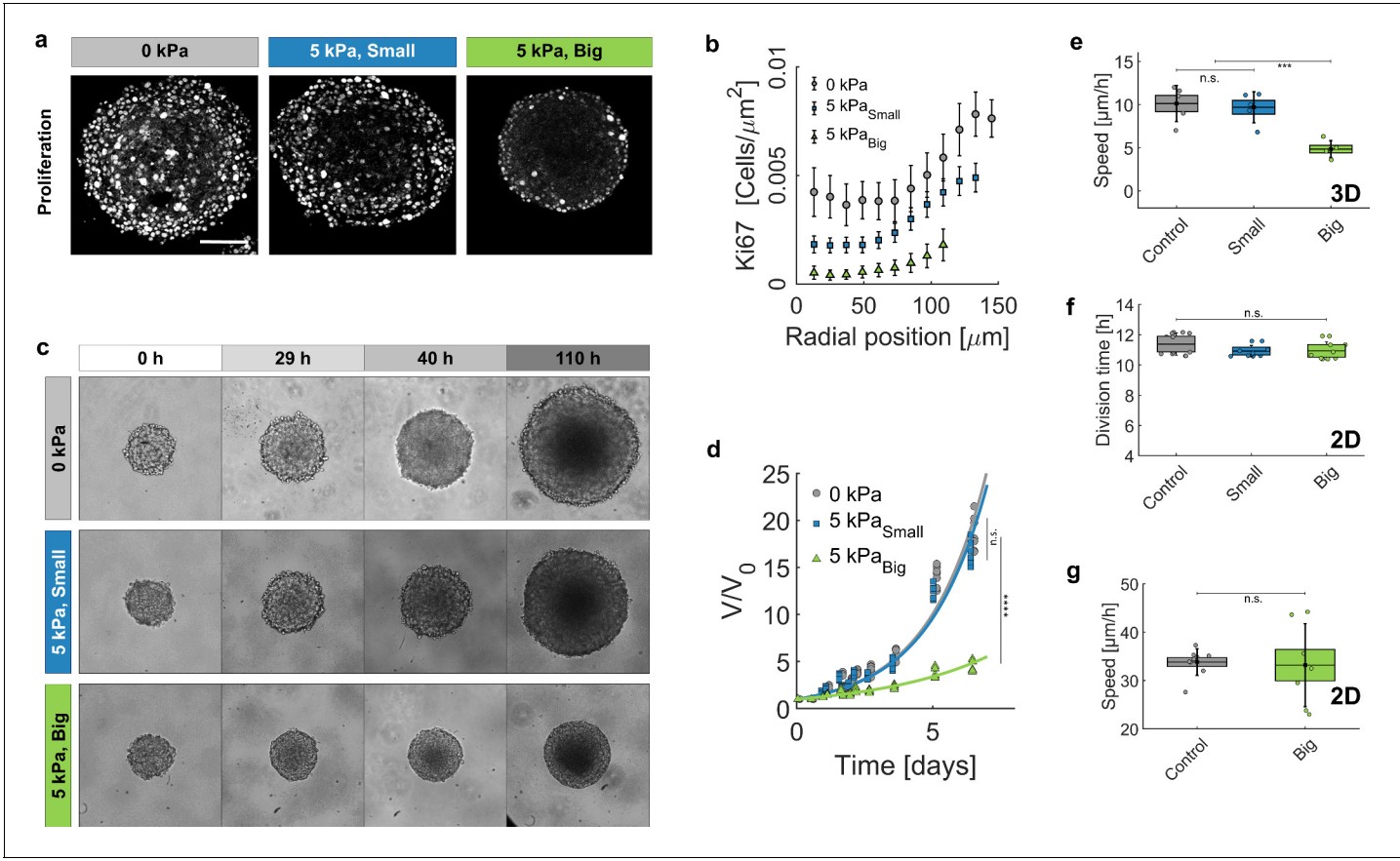

**Figure 4.** Growth of spheroids under pressure. (**a**) Proliferating cells inside MCS revealed by immunostaining of KI67 with no pressure, under global compression of $\Pi_d$ = 5 kPa (big dextran) and under selective compression of the cells by the same amount (small dextran). Scale bar: 100 $\mu m$ (**b**) Density of Ki67-positive cells with respect to the distance from the center of the aggregate. Three conditions: No pressure (9 MCS), $\Pi_d$ = 5 kPa with small dextran (26 MCS) and $\Pi_d$ = 5 kPa with big dextran (19 MCS). Error bars = standard error of the mean. (**c**) Time evolution of the spheroid sizes (Full images are 700 × 700 µm) and (**d**) quantification of the volume increase, in the three reference conditions. (**e**) Cell migration speed within MCS also significantly depends on ECM compression. N = 5 independent experiments per condition. Error bars represent ± SEM. Experiments were repeated at least on three independent samples. (**f**) Division time of CT26 cells in 2D (Petri dish), respectively with no pressure (11.4 ± 0.5 hr), with $\Pi_d$ = 5 kPa/small dextran (11.0 ± 0.3 hr) and with $\Pi_d$ = 5 kPa/big dextran (10.9 ± 0.4 hr). (**g**) Mean velocity of individual cells on a Petri dish, before (N = 9) and after compression (N = 7).

The online version of this article includes the following source data for figure 4:

**Source data 1.** Data for *Figure 4b*.
**Source data 2.** Data for *Figure 4d*.
**Source data 3.** Data for *Figure 4e*.
**Source data 4.** Data for *Figure 4f,g*.

pressure the density of Ki67-positive cells uniformly decreases across the MCS. Consequently, the ratio between the proliferating cells in the periphery of the MCS and those in its core increases under pressure: 2 without pressure, 2.5 under 5 kPa exerted by small dextran and 5 when the same pressure is exerted by big dextran. To quantify the change of cell division rate, we monitored the volumetric growth of the spheroid for three conditions (control, small and big dextran) and for several days (*Figure 4c–d*). In all cases, the spheroids initially grew exponentially (continuous lines). However, the MCS growth rate (time to double its volume) almost doubled under the big dextran compression, increasing from $36 \pm 1$ hr for the control and small dextran conditions (gray circles and blue squares, respectively) to $68 \pm 4$ hr for the compression with big dextran (green triangles).

Because experiments with MCS are typically performed in solution, where a metastatic behavior is not possible, we evaluated the cell motility within the aggregate, using the Dynamic Light Scattering technique introduced by *Brunel et al., 2020* (see details in Appendix 11). The mean migration velocity of cells was reduced by 50% at $\Pi_d = 5$ kPa with big dextran, as compared to the unstressed case (*Figure 4e*). Strikingly, both proliferation and motility remained almost unaltered when the MCS were exposed to an equivalent pressure ($\Pi_d = 5$ kPa) applied by small dextran to selectively compress the cells while leaving the native ECM unstrained (small Dextran, blue).

To verify that neither proliferation nor motility are directly modified by the direct action of dextran in contact with the cells, we measured the proliferation and the velocity of individual cells plated in a Petri dish. Measurements were performed at low density to permit cell proliferation and migration. The results (*Figure 4f-g*) show that both proliferation and motility remained similar, before and after the addition of dextran at a final pressure $\Pi_d \simeq 5$ kPa.

Since the interstitial space is dehydrated under osmotic compression, cells may get in contact with each other, occasioning contact inhibition of proliferation and locomotion. However, it is also possible that cells sense and react to the stress in the ECM. To discriminate between these two hypotheses, we embedded individual cells in a MG matrix, before compressing the whole system with an osmotic pressure $\Pi_d = 5$ kPa using either small or big dextran. After a few days, we observed two clearly different phenotypes. Cells grown without pressure or in the presence of small dextran were sparse in the MG (*Figure 5a*, left panel). Conversely, cells cultured with big dextran proliferated locally (*Figure 5a*, right panel). Therefore, MG compression appears to inhibit cell motility and to promote the formation of mini-spheroids, which suggests that ECM compression has a direct effect on the cell-ECM biophysical signaling. The different cell morphology is particularly clear in the organization of the actin cytoskeleton. Cytoplasmic actin labeling revealed the presence of numerous protrusions, associated with high cell anisotropy, in cells cultured in a relaxed MG matrix (*Figure 5b*, left and middle panels), whereas cells appeared smooth and formed round structures, when the MG was compressed (*Figure 5b*, right panel). Of note, cells at the MG surface often extended outside the MG. Those cells not fully embedded in MG were excluded from our analysis.

These different morphologies also correlates with different motilities. Cells embedded in a compressed MG were nearly immobile, while they migrated through relaxed MG with a velocity comparable to that measured on flat surfaces. The results are summarized in *Figure 5c*, where we report $\simeq 40$ trajectories per condition. To highlight differences and similarities between the three compression conditions, the starting points of all trajectories are translated to the origin and, although isotropic, they are divided in three quadrants. Quantification is reported in *Figure 5d*. From this experiment we conclude that whereas no appreciable differences are observable between control and the small dextran condition, the cell motility dramatically drops under MG compression with big dextran.

To quantify the effect of ECM compression on proliferation, we prepared several samples with the same number of hoechst-stained cells embedded in the MG and measured the overall fluorescence over time. *Figure 5e* shows the typical time evolution of the Hoechst signal for the three conditions: proliferation rate drops considerably when the MG is compressed (Big Dextran, $\triangle$), compared to the case without pressure ($o$), but also compared to the case where the pressure is selectively exerted on the cells with no MG compression (Small Dextran, $\square$). *Figure 5f* quantifies the mean growth rate, measured on at least 15 samples for each condition, collected on eight independent experiments (different days and cell passages). Under pressure, the matrigel get dehydrated and compacted, which may directly influence cell proliferation. On the one hand a denser matrigel is less compressible and, thus, less favorable to cell proliferation (*Baker et al., 2015*). On the other hand, matrigel compression concentrates matrix-bound growth factors, which may promote cell

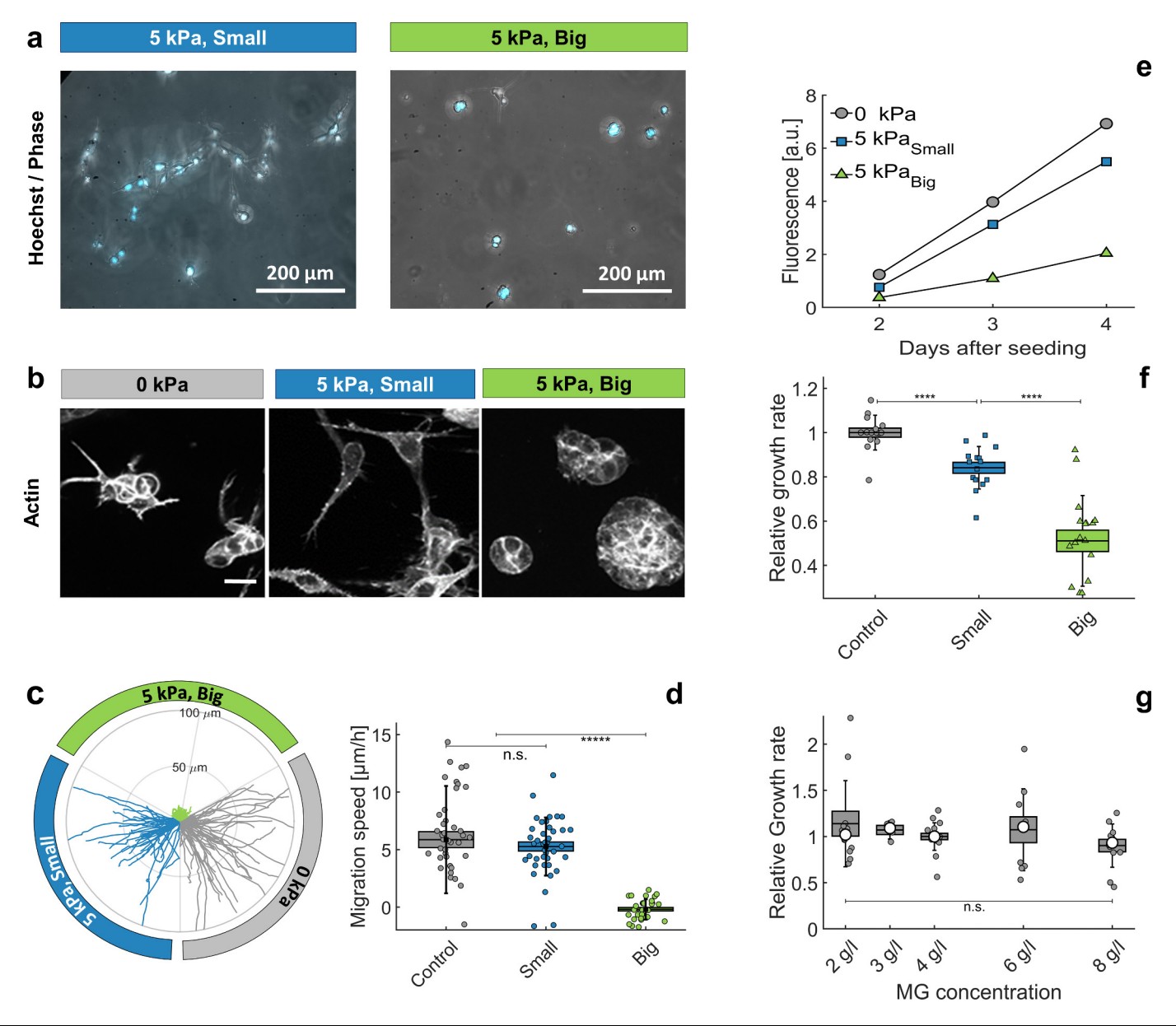

**Figure 5.** Individual cells in Matrigel. (**a**) Hoechst-labeled cell nuclei superimposed to phase image. Images are taken after 2 days of proliferation in MG, either with small (left panel) or big (right panel) dextran molecules. Maximal projection from epifluorescence stacks. (**b**) Cell morphology and anisotropy revealed by labeling of cytoplasmic actin. Maximal projection of 50 μ m confocal Z-stack. In relaxed MG, the cells appear more elongated and with long protrusions. (**c**) Cell motility in MG under different compression states. Starting points of trajectories are translated to the origin, to highlight the typical distance over which cells move in the three compressive states. (**d**) Quantification of in-plane velocity extracted from mean square displacements, under different compression conditions. With no pressure or with small dextran (5 kPa), the average velocities are respectively $5.8 \pm 0.8$ $\mu m$/ hr and $5.2 \pm 0.5$ $\mu m$/ hr. Under 5 kPa exerted by big dextran, the cells are immobile ($v = 0.5 \pm 0.4$ $\mu m$), where the error is due to tracking uncertainties. (**e**) Temporal evolution of nuclear fluorescence intergrated over the whole sample. No pressure (*o*), 5 kPa with small dextran ($\square$) and 5 kPa with big dextran ($\triangle$). (**f**) Cell proliferation rate in the three conditions. n = 15, from eight independent experiments. (**g**) Cell proliferation rate at different initial matrigel concentration, with no pressure. Boxes represent the mean values ± SEM, error bars correspond to the standard deviation, small markers are individual experiments and large markers the median.

The online version of this article includes the following source data for figure 5:

**Source data 1.** Data for *Figure 5c,d*.
**Source data 2.** Data for *Figure 5e,f,g*.

division. To determine which effect dominates, we measured the proliferation rate at different initial MG concentrations, between 2 g/l and 8 g/l (experiments reported in panels a-f were performed with MG at 4.5 g/l). Our experiments show that within this range, the matrigel density has little to no effect on cell proliferation (see *Figure 5g*). These experiments confirmed that it is the compressive stress transmitted to the cells by the surrounding ECM, rather than a direct osmotic pressure on the cells, that strongly impacts cell motility and proliferation.

## Discussion and conclusion

Large osmotic and mechanical pressures (of the order of 100 kPa) can cause a decrease in cell volume and consequently a deformation of the cell nucleus (*Zhou et al., 2009*; *Kim et al., 2015*) which may ultimately feedback on the cell proliferation. It has been recently proposed that the volume of the cell or its nucleus can be key to crucial processes such as proliferation, invasion, and differentiation *Guo et al., 2017*; *Han et al., 2020*. However the weak osmotic pressures (of the order of 1 kPa) that we apply have no measurable effect on the cell volume. In addition, it is well-known from a biological standpoint that such small volume perturbations are buffered by active regulatory processes in the cell (*Hoffmann et al., 2009*; *Cadart et al., 2019*). Yet, both cell proliferation and motility decrease in MCS submitted to weak osmotic compression. Our results show that, for such weak compressions, the cell volume is unchanged while the ECM located in between the cells is directly impacted. This mechano-sensitive role of the ECM could explain the reported evidences that osmotic pressures applied by big dextran and mechanical pressures similarly affect the growth of MCS. (*Helmlinger et al., 1997*; *Alessandri et al., 2013*; *Montel et al., 2011*). Indeed, in this case the osmotic pressure induces a mechanical one applied on the cells through the ECM drainage. Thanks to its bulk modulus $K_{ECM} \simeq 1$ kPa, the ECM behaves as a pressure sensor for the cell in the kPa range. Of note, stress relaxation in the ECM could occur through cleavage and remodeling of its components and such active processes should be quantified in the future.

Several mechanisms may explain how the dehydration of the extracellular matrix can result in an inhibition of proliferation and motility. First, the reduction of the interstitial space promotes interactions between neighbouring cells, which may activate contact inhibition signals of both proliferation and locomotion (*Roycroft and Mayor, 2016*). Second, the ECM porosity and tortuosity change within a compressed MCS, such that its effective permeability to oxygen, nutrients, growth factors and cytokines is reduced and might activate inhibition signals without cell-cell contact. However, both options are incompatible with the results we obtained with single cells embedded in MG (*Figure 5*). During the first 2–3 days after seeding, cells are either isolated or grouped in aggregates of two to four cells with limited cell-cell contacts. Additionally, a key factor limiting the diffusion of oxygen and nutrients in MCS is the tortuosity of the interstitial space (*Bläßle et al., 2018*). This constraint is simply absent in experiments with single cells embedded in MG, suggesting that the cell proliferation inhibition is most probably not related to hypoxia and starvation.

The present work therefore points at a direct mechanosensitive response of cells to the ECM deformation. The microscopic structure of the ECM is modified under compression (e.g. density increase and reduction of porosity), with consequences on the ECM rheology. Compression of the ECM is clearly accompanied by an increase in its bulk modulus and, due to its fibrillar structure, to a non-trivial and non-linear evolution of its stiffness (*Sopher et al., 2018*; *Kurniawan et al., 2016*). For example, the rheological properties of synthetic ECM have been shown to affect growth of aggregates and single cells through the regulation of streched-activated channels (*Nam et al., 2019*). As integrin-dependent signals and focal adhesion assembly are regulated by the stress and strain between the cell and the ECM, the osmotic compression may steer the fate of cells in terms of morphology, migration, and differentiation (*Pelham and Wang, 1997*; *Choquet et al., 1997*; *Sunyer et al., 2016*; *Isenberg et al., 2009*; *Butcher et al., 2009*; *Engler et al., 2006*; *Staunton et al., 2019*; *Panzetta et al., 2019*). This aspect is also relevant from an oncological point of view. Indeed the ECM is strongly modified in tumour tissues and the solid stress within tumors can reach several kPa, which is in accordance with the pressure applied here (*Nia et al., 2017*). For example, in tumors, there is a decrease in the ratio collagen/hyaluronan (*Voutouri et al., 2016*). The latter, more hydrophilic than the first one, tends to swell and stiffen the ECM. Whether a corrupted matrix is a contributing cause or the consequence of the neoplasia remains an open question, but

the correlation between matrix mechanics and uncontrolled proliferation is more and more widely accepted (*Bissell et al., 2002*; *Lelièvre and Bissell, 2006*; *Broders-Bondon et al., 2018*).

In future experiments it will be crucial to identify whether the ECM compression and the associated changes in stiffness play a dominant role, or if – as we suggest – the mechanical stress applied on the cell through the ECM is the key ingredient directly triggering the cell biological adaptation in term of proliferation and motility.

# Materials and methods

**Key resources table**

| Reagent type (species) or resource | Designation | Source or reference | Identifiers | Additional information |
|---|---|---|---|---|
| Cell Line (*mouse*) | CT26 | ATCC CRL-2638 | RRID:CVCL_7256 | |
| Chemical compound,drug | Matrigel/MG | Corning; 354234 | | |
| Chemical compound, drug | Small Dextran | Sigma Aldrich; D9260 | | |
| Chemical compound, drug | Big Dextran | Sigma-Aldrich; D5376 | | |
| Antibody | Anti-Fibronectin (*Monoclonal mouse*) | Sigma Aldrich; F7387 | RRID:AB_476988 | (1:200) |
| Antibody | Anti-Ki-67 (*Polyclonal rabbit*) | Sigma Aldrich; AB9260 | RRID:AB_2142366 | (1:200) |

## Cell culture, MCSs formation, and growth under mechanical stress

CT26 (mouse colon adenocarcinoma cells, ATCC CRL-2638); American Type Culture Collection were cultured under 37°C, 5% $CO_2$ in DMEM supplemented with 10% calf serum and 1% antibiotic/antimycotic (culture medium). Cells are texted every month for mycoplasma. None of the experiments was made using cells with mycoplasma. Spheroid were prepared on agarose cushion in 96 well plates at the concentration of 500 cell/well and centrifuged initially for 5 min at 800 rpm to accelerate aggregation. After 2 days, Dextran (molecular mass 1, 10, 40, 70, 100, 200, 500, and 2000 kDa; Sigma-Aldrich, St. Louis, MO) was added to the culture medium to exert mechanical stress, as previously described (*Monnier et al., 2016*). To follow spheroid growth over the time, phase contrast images were taken daily. Spheroid were kept under constant pressure over observation period. Images were analysed manually using Imagej. Each experiment was repeated three times, with 32 individual spheroids per condition.

## Measurement of MCSs volume

The area of the MCS equatorial section was measured before and after addition of dextran, then converted to volume assuming that the MCS is spherical. To induce compression, half of the culture medium was removed and replaced with fresh medium containing dextran 2X (two-fold the target concentration). The error affecting this measurement mainly comes from the fact that spheroids rotate during buffer exchange. As they are not perfectly spherical, the area of the equatorial section may change by up to few percent. To homogenize the experiments, control spheroids (no pressure) were also measured before and after buffer exchange. In the latter case, 50% of the culture medium was simply aspirated and replaced by fresh medium not supplemented with dextran.

## Fabrication of Matrigel beads

Matrigel beads (Matrigel Corning, Ref: 354234) were prepared using vortex method (*Dolega et al., 2017*). Oil phase of HFE-7500/PFPE-PEG (1.5% w/v) was cooled down to 4°C. For 400 µL of oil, 100 µL of Matrigel were added. Solution was vortexed at full speed for 20 s and subsequently kept at 37°C for 20 min for polymerization. Beads were eventually transferred to PBS phase by washing out the surfactant phase.

## Fluorescence eXclusion method (single-cell volume measurements)

Cell volume was obtained using Fluorescence Exclusion microscopy (*Cadart et al., 2017*; *Zlotek-Zlotkiewicz et al., 2015*). Briefly, cells were incubated in PDMS chips, with medium supplemented with a fluorescent dye that does not enter the cells. Cells thus excluded fluorescence, and one extracted cellular volume by integrating the fluorescence intensity over the whole cell . Chips for

volume measurements of single cells were made by pouring a mixture (1:10) of PMDS elastomer and curing agent (Sylgard 184) onto a brass master and cured at 80 °C for at least of 2 hr. Inlet and outlets were punched with a 3 mm biopsy puncher. Chips were prepared few days before, bounded with oxygen plasma for 30 s, warmed up at 80°C for 3 min then incubated with Poly-l-lysine (sigma) for 30 min to 1 hr, washed with PBS, then washed with $dH_2O$, dried and stored sealed with a paraffin film. The chambers were washed with PBS before cell injection. Imaging started within 10 min after cell injection in order to prevent adhesion and thus cells response to the shear stress generated by the medium exchange. Acquisition was performed at 37°C in $CO_2$ independent medium (Life Technologies) supplemented with 1 g/L FITC dextran (10 kDa, from Sigma Aldrich) on an epifluorescence microscope (Leica DMi8) with a 10x objective (NA. 0.3 from LEICA). Master molds were fabricated on a brass substrate with a micromilling machine (MiniMill/3; Minitech) using a 100-µm-diameter milling cutter (Minitech). Height profiles and surface roughness were measured with a vertical scanning interferometric profilometer (Brucker). 3D mold design and tool paths were generated using Autodesk Inventor Professional software (Autodesk). Molds for spheroid confinement were made with classical soft lithography techniques.

## Tissue compression experiments

Spheroids were harvested 4 or 5 days after cell seeding and injected in the *2D confiner* microsystem (*Figure 3a*) using a MFCS pressure controller (Fluigent). Spheroid were partially flattened between two parallel surfaces, perpendicular to the optical axis of the microscope, and rested for two to 5 hr to relax in the microsystem at 37°C in $CO_2$ independent medium. Before acquisition, medium supplemented with 2 g/L FITC-dextran (10 kDa from Sigma Aldrich) was injected to label the intercellular space. Medium exchange was performed manually using large inlets (<1 mm) during two-photon acquisition. Acquisitions were performed at 37°C on a Nikon C1 two-photon microscope coupled with a femtosecond laser at 780 nm with a 40x water-immersion (NA. 1.10) objective (Nikon). The *2D confiner* chip was made by pouring PDMS elastomere and curing agent (1:10) into the mold and cured for at least 2 hr. The chips were bounded to glass coverslips with 30 s oxygen plasma, immediately after bounding. A solution of PLL-g-PEG (Surface Solutions) at 1 g/l was injected and incubated for 30 min in humid atmosphere to prevent cell surface adhesion during the experiment. The chips were washed with $dH_2O$ and dried and sealed with a paraffin film. Fluorescence of the Intercellular space (ICS) was measured using MatLab software. As control and dextran solutions have different levels of fluorescence, the fluorescence in the ICS was normalized by the one outside the ICS in order to compensate these variations. Then, first the tissue (cells and ICS) was segmented with a thresholding procedure. The threshold was determined in order to obtained accurate segmentation of the ICS before application of the osmotic stress. The surface of the ICS was computed as the ratio of pixels in the ICS to the number of pixels of the tissue. For each spheroid, 50 planes - 13 µm above and below the equatorial plane - were taken into account to compute the change of the ICS surface.

## Cell culture in Matrigel

Experiments have been conceived to start the culture from individual cells embedded in Matrigel. At day 1, the cells were resuspended, then dispersed in a solution containing matrigel at the final concentration of 4.5 g/l. The cells were diluted to 10,000–50,000 cells/ml, a concentration at which the average distance between neighboring cells is about 250–400 µm. We therefore consider them as isolated entities. The MG/cell ensemble was gelified in 200 µl wells, at 37°C, for 30 min. To avoid cell sedimentation, we gently flipped the sample over, every two minutes. The samples were then redeposited in the incubator under three pressure conditions: no pressure and 5 kPa exerted by small and 5 kPa exerted big dextran.

## Cells migration in Matrigel

To quantify cell migration in Matrigel, individual cells were observed by phase contrast microscopy. Z-stacks were collected every 20 min and for several days, with slices spaced by 50 µm. Then the full stack was projected to one single layer (maximum intensity projection). Cells were tracked manually in the 2D plane, using the ImageJ MTrackJ plugin (https://imagescience.org/meijering/software/mtrackj/).

## Cryosectioning and immunostaining

Spheroids were fixed with 5% formalin (Sigma Aldrich, HT501128) in PBS for 30 min and washed once with PBS. For cryopreservation spheroids were exposed to sucrose at 10% (w/v) for 1 hr, 20% (w/v) for 1 hr and 30% (w/v) overnight at 4°C. Subsequently spheroids were transferred to a plastic reservoir and covered with Tisse TEK OCT (Sakura) in an isopropanol/dry ice bath. Solidified samples were brought to the cryotome (Leica CM3000) and sectioned into 15 μm slices. Cut layers were deposited onto poly-L-lysine coated glass slides (Sigma) and the region of interest was delineated with DAKO pen. Samples were stored at −20°C prior immunolabelling. For fibronectin and Ki67 staining samples were permeabilized with Triton X 0.5% in TBS (Sigma T8787) for 15 min at RT. Non-specific sites were blocked with 3% BSA (Bovine serum Albumin) for 1 hr. Then, samples were incubated with first antibody (Fibronectin, Sigma F7387, 1/200 and Ki67; Millipore ab9260, 1/500) overnight at 4°C. Subsequently samples were thoroughly washed with TBS three times, for 15 min each. A second fluorescent antibody (goat anti-mouse Cy3, Invitrogen; 1/1000) was incubated for 40 min along with phalloidin (1/500, Alexa Fluor 488, Thermo Fisher Scientific). After extensive washing with TBS (four washes of 15 min) glass cover slides were mounted on the glass slides with a Progold mounting medium overnight (Life Technologies P36965) and stored at 4°C before imaging.

## Statistical analysis

Student's t-test (unpaired, two tailed, equal variances) was used to calculate statistical significance as appropriate by using the *ttest2()* function of Matlab (MathWorks). Statistical significance is given by $*p<0.05$; $**p<0.01$; $***p<0.001$; $****p<0.0001$.

## Acknowledgements

We warmly thank J Prost and F Jülicher for drawing our attention to the potential impact of the poroelasticity in MCS, A Dawid and J Revilloud for the valuable suggestion to set up the proliferation assay in MG, C Verdier for the valuable exchanges about the evolution of the ECM rheology under stress, and T Boudou for the many fruitful discussion and careful proofreading of the manuscript. This work was supported by the Agence Nationale pour la Recherche (Grant ANR-13-BSV5-0008-01), by the Institut National de la Santé et de la Recherche Médicale (Grant PC201407), by the Centre National de la Recherche Scientifique (Grant MechanoBio 2018), by the Comité de Haute-Savoie de la Ligue contre le Cancer, and by a CNRS Momentum grant (PR).

## Additional information

### Funding

| Funder | Grant reference number | Author |
|---|---|---|
| Agence Nationale de la Recherche | ANR-13-BSV5-0008-01 | Giovanni Cappello |
| Centre National de la Recherche Scientifique | MechanoBio 2018 | Giovanni Cappello |
| Ligue Contre le Cancer | | Sylvain Monnier |
| Institut National de la Santé et de la Recherche Médicale | PC201407 | Giovanni Cappello |
| Centre National de la Recherche Scientifique | | Pierre Recho |

The funders had no role in study design, data collection and interpretation, or the decision to submit the work for publication.

### Author contributions

Monika E Dolega, Conceptualization, Data curation, Validation, Investigation, Writing - original draft; Sylvain Monnier, Data curation, Investigation, Writing - review and editing; Benjamin Brunel, Investigation; Jean-François Joanny, Formal analysis, Writing - review and editing; Pierre Recho,

Conceptualization, Formal analysis, Methodology, Writing - original draft; Giovanni Cappello, Conceptualization, Data curation, Formal analysis, Funding acquisition, Validation, Investigation, Methodology, Writing - original draft, Project administration

### Author ORCIDs
Benjamin Brunel http://orcid.org/0000-0003-2858-5074
Giovanni Cappello https://orcid.org/0000-0002-5012-367X

### Decision letter and Author response
Decision letter https://doi.org/10.7554/eLife.63258.sa1
Author response https://doi.org/10.7554/eLife.63258.sa2

## Additional files
### Supplementary files
• Transparent reporting form

### Data availability
Data relating to figures 2, 3, 4, 5 and appendix B are available at https://osf.io/n6ra2/?view_only=059da2ebcd064b75bd12c0c2008b9a6a.

The following dataset was generated:

| Author(s) | Year | Dataset title | Dataset URL | Database and Identifier |
|---|---|---|---|---|
| Cappello G | 2021 | Extracellular matrix in multicellular aggregates | https://osf.io/n6ra2/?view_only=059da2ebcd064b75b-d12c0c2008b9a6a | Open Science Framework, n6ra2 |

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

# Appendix 1

## A Theoretical model of the osmotic compression of a single cell nested in matrigel

Our aim is to qualitatively understand the nature of the steady state mechanical stress and displacement of a cell nested in a matrix in two paradigmatic situations:

- When some small osmolytes (typically dextran) that can permeate the matrix pores are introduced in the solution,
- When some big osmolytes that are excluded from the matrix are introduced in the solution.

The matrix is a meshwork of biopolymers permeated by an aqueous solution containing ions. These ions can also permeate the cell cytoplasm via specific channels and pumps integrated in the plasmic membrane (*Hoffmann et al., 2009*; *Lang et al., 1998*). For simplicity, we restrict our theoretical description to $Na^+$, $K^+$, and $Cl^-$ ions which have specific channels and a well studied pump (*Therien and Blostein, 2000*) which actively pumps out three sodium ions in exchange of having two potassium ions in. Attached right under the cell membrane via some specific cross-linkers (*Diz-Muñoz et al., 2010*), the cell cortex is a thin 'muscle-like' actin network cross-linked by passive and contractile cross-linkers such as myosin II. The cortex has been shown to be an important regulator of the cell surface tension (*Clark and Paluch, 2011*; *Salbreux et al., 2012*) as exemplified during motility (*Hawkins et al., 2011*; *Farutin et al., 2019*) and cell morphogenesis (*Turlier et al., 2014*; *Sedzinski et al., 2011*; *Tinevez et al., 2009*; *Charras et al., 2008*). The cell membrane and cortex enclose the cytoplasm a meshwork of macromolecules permeated by water and containing the aforementioned ions. See *Appendix 1—figure 6* for a scheme of the model.

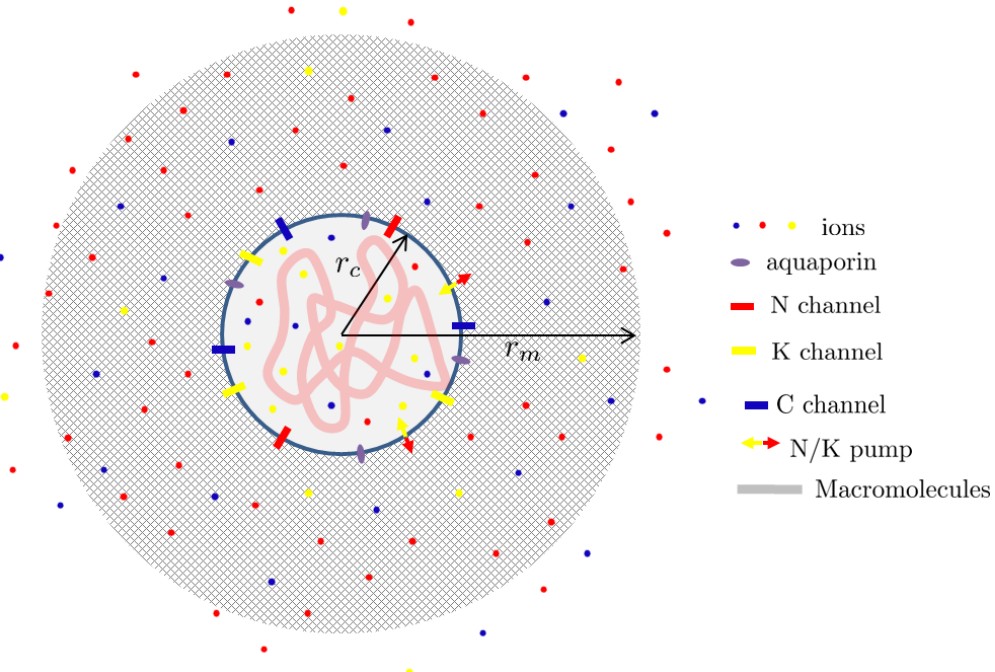

**Appendix 1—figure 1.** Scheme of a cell nested in a porous matrix.

For simplicity we assume a spherical geometry with a cell of radius $r_c$ inside a matrix ball of radius $r_m$. Each point in the space $\mathbf{x}$ can therefore be localized by its radial position $\mathbf{x} = r\mathbf{e}_r$ where $\mathbf{e}_r$ is radial unit vector. We assume a spherical symmetry of the problem such that all the introduced physical fields are independent of the angular coordinates $\theta$ and $\varphi$. Throughout this text, we restrict ourselves to a linear theory which typically holds when the deformation in the matrix is assumed to remain sufficiently small. A more quantitative theory would require to take into account both the non-linear aspects of the matrix deformation and the osmotic pressure created by the polymer.

## A.1 Conservation laws at the cell-matrix interface

### Water conservation

From Kedem-Katchalsky theory (*Staverman, 1952*; *Kedem and Katchalsky, 1958*; *Kedem and Katchalsky, 1963*; *Baranowski, 1991*; *Elmoazzen et al., 2009*), assuming that the aquaeous solvent moves through specific and passive channels, the aquaporins (*Day et al., 2014*), we can express the incoming water flux $\mathbf{j}_w$ in the cell at $r = r_c$ as (*Yi et al., 2003*; *Hui et al., 2014*; *Strange, 1993*; *Hoffmann et al., 2009*; *Mori, 2012*; *Cadart et al., 2019*):

$$\mathbf{j}_w.\mathbf{e}_r = L_p[p_m - p_c - (\Pi_m - \Pi_c)], \tag{4}$$

where $\Pi_{m,c}$ denote the osmotic pressures in the matrix phase and the cell while $p_{m,c}$ are the hydrostatic pressures defined with respect to the external (i.e. atmospheric) pressure. The so-called filtration coefficient $L_p$ is related to the permeability of aquaporins. In a dilute approximation which we again assume for simplicity, the osmotic pressure is dominated by the small molecules in solution and thus reads

$$\Pi_m = k_B T(N_m + K_m + C_m + D_m) \text{ and } \Pi_c = k_B T(N_c + K_c + C_c), \tag{5}$$

where $k_B$ is the Boltzmann constant, $T$ the temperature, $N_{c,m}$, $K_{c,m}$ and $C_{c,m}$ are the (number) concentrations of sodium, potassium and chloride in the cytoplasm and the extra-cellular medium and $D_m$ is the extra-cellular Dextran (necessary small as big are excluded) concentration in the matrix phase. We neglect in (*Equation 5*) the osmotic contribution associated with the large macromolecules composing the cell organelles and the cytoskeleton compared to the ionic contributions. In a similar manner, the osmotic contribution of the matrix polymer is also neglected. At steady state, the water flux vanishes ($\mathbf{j}_w = 0$) leading to the relation at $r = r_c$,

$$p_m - p_c = \Pi_m - \Pi_c. \tag{6}$$

### Ions conservation

As each ion travels through the plasma membrane via specific channels and pumps, the intensities of each ionic current at $r = r_c$ is given by Nernst-Planck laws (*Mori, 2012*),

$$\begin{aligned}
i_N &= g_N\left[v_- \frac{k_B T}{q}\log\left(\frac{N_m}{N}\right)\right] + 3qj_p \\
i_K &= g_K\left[v_- \frac{k_B T}{q}\log\left(\frac{K_m}{K}\right)\right] - 2qj_p \\
i_C &= g_C\left[v_+ \frac{k_B T}{q}\log\left(\frac{C_m}{C}\right)\right],
\end{aligned} \tag{7}$$

where $g_{N,K,C}$ are the respective conductivities of ions, $v_c$ is the cell membrane potential, $q$ is the elementary charge and $j_p$ is the pumping rate associated to the Na-K pump on the membrane which is playing a fundamental role for cellular volume control (*Hoffmann et al., 2009*). The factors 3 and 2 are related to the stochiometry of the sodium potassium pump. Again, in steady state, currents $i_{N,K,C} = 0$, leading to the Gibbs-Donnan equilibrium:

$$N_c = N_m e^{-\frac{q(v_c - v_N)}{k_B T}}, K_c = K_m e^{-\frac{q(v_c - v_K)}{k_B T}} \text{ and } C_c = C_m e^{\frac{qv_c}{k_B T}}, \tag{8}$$

where the active potentials related to the pumping activity $v_{N,K}$ are $v_N = -3qj_p/g_N$ and $v_K = 2qj_p/g_K$.

Supposing that the cell membrane capacitance is vanishingly small (*Mori, 2012*), we can neglect the presence of surface charges and impose an electro-neutrality constraint for the intra-cellular medium:

$$N_- C_+ K_- \rho_z = 0, \tag{9}$$

where $z_c$ is the average number of (negative in the physiological $\mathrm{pH} = 7.4$ conditions) electric charges carried by macromolecules inside the cell and $\rho_c$ is their density. As macromolecules are trapped inside the cell membrane, we can express $\rho_c = X_c/(4\pi r_c^3/3)$ where $X_c$ is the number of macro-

molecules which is fixed at short timescale and only increases slowly through synthesis as the amount of dry mass doubles during the cell cycle (*Cadart et al., 2019*).

## Force balance

At the interface between the cell and the matrix $(r = r_c)$, we can express the mechanical balance as

$$\mathbf{\Sigma}_c^{\text{bulk}}\mathbf{e}_r + \sigma_c^{\text{surf}}\mathbf{e}_r = \mathbf{\Sigma}_m\mathbf{e}_r. \tag{10}$$

In (*Equation 10*), $\mathbf{\Sigma}_c^{\text{bulk}}$ is the Cauchy stress in the cytoplasm which we decompose into $\mathbf{\Sigma}_c^{\text{bulk}} = \mathbf{\Sigma}_c^{\text{skel}} - p_c\mathbf{I}$, with a first contribution due to the cytoskeleton and a second contribution due to the hydrostatic pressure in the cytosol. The identity matrix is denoted I. The contribution due to the mechanical resistance of the cortex and membrane is denoted $\sigma_c^{\text{surf}}$. In our spherical geometry, we can express $\sigma_c^{\text{surf}} = 2\gamma_c/r_c$ where $\gamma_c$ is a surface tension in the cell contour. Finally $\mathbf{\Sigma}_m$ is the stress in the matrix phase for which we postulate a poro-elastic behavior such that, $\mathbf{\Sigma}_m = \mathbf{\Sigma}_m^{\text{el}}(\epsilon_m) - p_m\mathbf{I}$ (the Biot coefficient [*Coussy, 2004*] is assumed to be one). where

$$\mathbf{\Sigma}_m^{\text{el}} = 2G\mathbf{E}_m + \left(K_d - \frac{2G}{3}\right)\text{tr}(\mathbf{E}_m)\mathbf{I}, \tag{11}$$

is the Hooke's law with $\mathbf{E}_m$ the (small) elastic strain in the matrix, $G$ the shear modulus and $K_d$ the drained bulk modulus.

In the absence of cytoskeleton and external matrix (*Equation 10*) reduces to Laplace law:

$$\frac{2\gamma_c}{r_c} = p_c - p_m$$

and more generally reads,

$$(\mathbf{\Sigma}_c^{\text{skel}} - \mathbf{\Sigma}_m^{\text{el}})\mathbf{e}_r.\mathbf{e}_r + \frac{2\gamma_c}{r_c} = p_c - p_m. \tag{12}$$

Such relation provides the hydrostatic pressure jump at the cell membrane $(r = r_c)$ entering in the osmotic balance (*Equation 6*) and, combining (*Equation 6*) and (*Equation 12*), we obtain

$$(\mathbf{\Sigma}_c^{\text{skel}} - \mathbf{\Sigma}_m^{\text{el}})\mathbf{e}_r.\mathbf{e}_r + \frac{2\gamma_c}{r_c} = \Pi_c - \Pi_m. \tag{13}$$

## Conservation laws in the extracellular matrix

### Water conservation

Assuming that the extracellular fluid follows a Darcy law, mass conservation of the incompressible water permeating the matrix can be expressed as

$$\frac{\partial n}{\partial t} - \frac{\kappa}{\mu}\frac{1}{r}\frac{\partial}{\partial r}\left(r\frac{\partial p_m}{\partial r}\right) = 0, \tag{14}$$

where $n$ is the matrix porosity, $\kappa$ the matrix permeability and $\mu$ the fluid viscosity. At steady state, $\partial_t n = 0$ and (*Equation 14*) is associated with no flux boundary conditions at $r_c$ and $r_m$ given by

$$\frac{\partial p_m}{\partial r}\Big|_{r_m, r_c} = 0.$$

It follows that $p_m$ is homogeneous in the matrix and its value is imposed by a relation similar to (*Equation 6*) with an infinitely permeable membrane at $r_m$ :

$$p_m(r) = \Pi_m - \Pi_e. \tag{15}$$

In (*Equation 15*), $\Pi_e$ is the external osmotic pressure which reads

$$\Pi_e = k_B T(N_e + K_e + C_e + D_e) \tag{16}$$

where $N_e$, $K_e$ and $C_e$ denote the ions concentrations in the external solution and $D_e$ the concentration of Dextran added to the external solution.

## Ions conservation

As we are interested in the steady-state only, the Poisson-Nernst fluxes of ions concentrations in the matrix locally vanish leading to:

$$\frac{\partial N_m}{\partial r} + \frac{N_m q}{k_B T}\frac{\partial v_m}{\partial r} = \frac{\partial K_m}{\partial r} + \frac{K_m q}{k_B T}\frac{\partial v_m}{\partial r} = \frac{\partial C_m}{\partial r} - \frac{C_m q}{k_B T}\frac{\partial v_m}{\partial r} = 0,$$

where $v_m(r)$ is the electro-static potential in the matrix.

As $v_m$ is defined up to an additive constant, we chose that $v_m(r_m) = 0$ and, imposing the continuity of ions concentrations at the transition between the matrix and the external solution $N_m|_{r_m} = N_e$, $K_m|_{r_m} = K_e$ and $C_m|_{r_m} = C_e$, we obtain

$$N_m = N_e e^{-\frac{q v_m}{k_B T}}, K_m = K_e e^{-\frac{q v_m}{k_B T}} \text{ and } C_m = C_e e^{\frac{q v_m}{k_B T}}. \tag{17}$$

Next, we again suppose for simplicity that the capacitance of both the porous matrix and the external media are vanishingly small leading to the electro-neutrality constraints

$$\begin{aligned} N_m + K_m - C_m - z_m \rho_m &= 0 \\ N_e + K_e - C_e &= 0, \end{aligned} \tag{18}$$

where $z_m$ is the number of negative charges carried by the biopolymer chains forming the matrix and $\rho_m$ is their density. As we use uncharged Dextran, its concentration does not enter in expressions (*Equation 18*). Using, (*Equation 17*) in tandem with (*Equation 18*), we obtain

$$v_m = -\frac{k_B T}{q}\sinh^{-1}\left(\frac{z_m \rho_m}{2 C_e}\right). \tag{19}$$

Re-injecting this expression into (*Equation 17*), we obtain the steady state concentrations of ions in the matrix phase:

$$N_m = N_e e^{\sinh^{-1}\left(\frac{z_m \rho_m}{2 C_e}\right)}, K_m = K_e e^{\sinh^{-1}\left(\frac{z_m \rho_m}{2 C_e}\right)} \text{ and } C_m = C_e e^{-\sinh^{-1}\left(\frac{z_m \rho_m}{2 C_e}\right)}. \tag{20}$$

Next, we make the realistic assumption that the chloride concentration (number of ions per unit volume) is much larger than the density of fixed charges carried by the polymer chains (number charges per unit volume): $z_m \rho_m / C_e \ll 1$. Indeed using the rough estimates of Section 'Cell volume in the reference situation', the average number of charge carried per amino acid is 0.06 and the typical concentration of matrix is 5 g/l. As the molar mass of an amino acid is roughly 150 g/mol, we can estimate in moles that $z_m \rho_m \simeq 2$ mM while $C_e \simeq 100$ mM. We can thus simplify (*Equation 20*) up to first order to obtain,

$$N_m = N_e\left(1 + \frac{z_m \rho_m}{2 C_e}\right), K_m = K_e\left(1 + \frac{z_m \rho_m}{2 C_e}\right) \text{ and } C_m = C_e\left(1 - \frac{z_m \rho_m}{2 C_e}\right). \tag{21}$$

As a result, we obtain that the only steady state contribution of

$$\Pi \overset{\text{def}}{=} \Pi_e - \Pi_m = k_B T(D_e - D_m) = \begin{cases} 0 & \text{for small Dextran} \\ k_B T D_e & \text{for big Dextran,} \end{cases} \tag{22}$$

is imposed by Dextran since the ions only start to contribute to this difference at second order in the small parameter $z_m \rho_m / C_e$. We therefore conclude that, in good approximation, $\Pi$ vanishes for small Dextran molecules that can permeate the matrix and equates to the imposed and known quantity $k_B T D_e$ for big Dextran molecules that cannot enter the matrix pores.

It then follows from *Equation 15* that the hydrostatic pressure equilibrates with the imposed osmotic pressure,

$$p_m(r) = -\Pi.$$ (23)

## Force balance

Using the spherical symmetry of the problem, the only non vanishing components of the stress tensor are $\boldsymbol{\Sigma}_m^{rr}$ and $\boldsymbol{\Sigma}_m^{\theta\theta} = \boldsymbol{\Sigma}_m^{\varphi\varphi}$. Therefore, the local stress balance reads

$$\frac{\partial \boldsymbol{\Sigma}_m^{rr}}{\partial r} + \frac{2}{r}(\boldsymbol{\Sigma}_m^{rr} - \boldsymbol{\Sigma}_m^{\theta\theta}) = 0,$$

Assuming a small enough displacement, the non-vanishing components of the strain tensor are given by, $\mathbf{E}_m^{rr} = \partial u_r / \partial r$ and $\mathbf{E}_m^{\theta\theta} = \mathbf{E}_m^{\varphi\varphi} = u_r / r$ where $u_r$ is the radial (and only non-vanishing) displacement component from an homogeneous reference configuration corresponding to a situation where the matrix is not subjected to any external loading and $r_{c,m} = R_{c,m}$. Using the poro-elastic constitutive behavior (*Equation 11*), $u_r$ satisfies

$$\left(K_d + \frac{4}{3}G\right)\left(\frac{\partial^2 u_r}{\partial r^2} + \frac{2}{r}\frac{\partial u_r}{\partial r} - \frac{2}{r^2}u_r\right) = \frac{\partial p_m}{\partial r}.$$ (24)

This equation is supplemented with the traction free boundary condition at $r = r_m$

$$\boldsymbol{\Sigma}_m \mathbf{e}_r = 0.$$ (25)

Combined with (*Equation 23*), the two above *Equations (24) and (25)* lead to the solution

$$u_r(r) = \epsilon^0 r + \frac{r_m^3(\Pi + 3K_d\epsilon^0)}{4Gr^2},$$ (26)

where the introduced constants $\epsilon^0$ is found using the displacement continuity at the cell matrix-interface:

$$u_r(r_c) = u \overset{\text{def}}{=} r_c - R_c,$$ (27)

with $u$ given by the change of the cell radius from a reference configuration with radius $R_c$. The general expression of $u_r$ therefore reads,

$$u_r(r) = \frac{ur_c^2(4Gr^3 + 3K_d r_m^3) + \Pi r_m^3(r_c^3 - r^3)}{r^2(4Gr_c^3 + 3K_d r_m^3)},$$ (28)

leading to the following form of the total mechanical stress in the surrounding matrix:

$$\boldsymbol{\Sigma}_m(r) = \frac{2Gr_c^2(3K_d u + r_c\Pi)}{r^3(4Gr_c^3 + 3K_d r_m^3)} \times \begin{pmatrix} 2(r^3 - r_m^3) & 0 & 0 \\ 0 & (2r^3 + r_m^3) & 0 \\ 0 & 0 & (2r^3 + r_m^3) \end{pmatrix}.$$ (29)

## A.3 Formulation of the model

Combining (*Equation 5*) with (*Equation 13*) and taking into account (*Equation 21*), we obtain the relation linking the cell mechanics and the osmotic pressures inside the cell and outside the matrix:

$$(\boldsymbol{\Sigma}_c^{\text{skel}} - \boldsymbol{\Sigma}_m^{\text{el}})\mathbf{e}_r \cdot \mathbf{e}_r + \frac{2\gamma_c}{r_c} = k_B T(N_c + K_c + C_c - N_e - K_e - C_e - D_m).$$

We suppose that the stress in the cytoskeleton is regulated at a homeostatic tension such that $\boldsymbol{\Sigma}_c^{\text{skel}} \mathbf{e}_r \cdot \mathbf{e}_r \overset{\text{def}}{=} \Sigma_a$ is a fixed given constant modeling the spontaneous cell contractility. We can then linearize the cell mechanical contributions close to $r_c = R_c$ to obtain

$$\mathbf{\Sigma}_c^{\text{skel}}.\mathbf{e}_r.\mathbf{e}_r + \frac{2\gamma_c}{r_c} = \tilde{\Sigma}_a - k_c u,$$

where $\tilde{\Sigma}_a = \Sigma_a + 2\gamma_c/R_c$ and the effective cell mechanical stiffness is $k_c = 2\gamma_c/R_c^2$.

Using (*Equation 23*) and (*Equation 29*) close to $r_{c,m} = R_{c,m}$ we can express,

$$-\mathbf{\Sigma}_m^{\text{el}}\mathbf{e}_r.\mathbf{e}_r = \frac{12GK_d(R_m^3 - R_c^3)u + (4G + 3K_d)R_m^3 R_c \Pi}{4GR_c^4 + 3K_d R_c R_m^3}.$$

We therefore finally get the linear relation,

$$\tilde{\Sigma}_a + \tilde{k}u + \tilde{\alpha}\Pi = k_B T(N_c + K_c + C_c - N_e - K_e - C_e - D_m), \tag{30}$$

where,

$$\tilde{k} = -k_c + \frac{12GK_d(R_m^3 - R_c^3)}{4GR_c^4 + 3K_d R_c R_m^3} \text{and} \tilde{\alpha} = \frac{(4G + 3K_d)R_m^3}{4GR_c^3 + 3K_d R_m^3}.$$

In the limit where $R_m \gg R_c$,

$$\tilde{k} = -k_c + \frac{4G}{3R_c} \text{and} \tilde{\alpha} = 1 + \frac{4G}{3K_d}.$$

Next, using (*Equation 8*) and (*Equation 21*) and neglecting $z_m\rho_m/C_e \ll 1$ we obtain the relation linking the externally controlled osmolarity with the cell and matrix mechanics:

$$\frac{\tilde{\Sigma}_a + \tilde{k}u + (\tilde{\alpha}-1)\Pi}{k_B T} = N_e\left(e^{-\frac{q(v-v_N)}{k_B T}} - 1\right) + K_e\left(e^{-\frac{q(v-v_K)}{k_B T}} - 1\right) + C_e\left(e^{\frac{qv}{k_B T}} - 1\right) - D_e. \tag{31}$$

In a similar way, we combine (*Equation 8*) with (*Equation 9*) with again (*Equation 21*) in the limit where $z_m\rho_m/C_e \ll 1$ to express the electro-neutrality condition

$$N_e e^{-\frac{q(v-v_N)}{k_B T}} + K_e e^{-\frac{q(v-v_K)}{k_B T}} - C_e e^{\frac{qv}{k_B T}} = \frac{3z_c X_c}{4\pi R_c^3}\left(1 - \frac{3u}{R_c}\right), \tag{32}$$

where we have additionally linearized the right handside close to $r_c = R_c$.

The two *Equations (31) and (32)* constitute our final model.

## A.4 Cell volume in the reference situation

We begin by computing the cell radius and the cell membrane potential in the reference configuration where by definition $u = 0$ and $\Pi = D_e = 0$ as no Dextran is present at all. In this case, we solve for the membrane potential $v_c \overset{\text{def}}{=} V_c$ and radius $R_c$ in (*Equation 31*) and (*Equation 32*) to find their reference values. This computation strictly follows *Hoppensteadt and Peskin, 2012*.

Defining the non-dimensional parameters,

$$\beta = \frac{N_e e^{qv_N/(k_B T)} + K_e e^{qv_K/(k_B T)}}{C_e} \text{ and } \sigma = \frac{\tilde{\Sigma}_a}{k_B T C_e}$$

we find the reference radius and membrane potential,

$$R_c = \left(\frac{3z_c X_c}{4\pi C_e \sqrt{(\sigma+2)^2 - 4\beta}}\right)^{1/3}$$

and

$$V_c = \frac{k_B T}{2q}\log\left(-\sqrt{(\sigma+2)^2 - 4\beta} + \sigma + 2\right).$$

Given that the typical concentration of chloride ions outside the cell is of the order of 100

milimolar, the osmotic pressure $k_B T C_e$ is of the order $10^5 \mathrm{Pa}$ (i.e. an atmosphere). In sharp contrast, the typical mechanical stresses in the cytoskeleton and the cortex are of the order of $10^2 - 10^3 \mathrm{Pa}$(*Julicher et al., 2007*). Therefore, the non-dimensional parameter $\sigma$ is of the order of $\sigma \sim 10^{-3}$ and will be neglected in the following. We then finally obtain the reference values,

$$R_c = \left(\frac{3z_c X_c}{8\pi C_e \sqrt{1-\beta}}\right)^{1/3}, V_c = \frac{k_B T}{q}\log\left(1 - \sqrt{1-\beta}\right).$$

The pumping rate enables the cell to maintain a finite a volume. When $j_p \to 0$, $\beta \to 1$ and the cell swells to infinity because nothing balances the osmotic pressure due to the macromolecules trapped inside. So it is expected that dead cells will swell and lyse. The same happens if the pumping rate is to high. Indeed, as the membrane permeability of potassium is higher than the one of sodium, if the pumping rate is very high, a lot of potassium ions will be brought in (more than sodium ions will be expelled out) and to equilibrate osmolarity with the exterior, water will swell the cell until it bursts. Between these to unphysiological situations, computing the variation of volume with respect to the pumping rate, one gets that this variation vanishes when,

$$j_p^{opt} = \frac{k_B T}{q^2}\frac{g_N g_K}{g_N + g_K}\log\left(\frac{N_e g_K}{K_e g_N}\right).$$

At such pumping rate, the volume is less sensitive to small variations in the pumping rate that may occur.

## Rough estimates

The computation of the effective charge carried by macromolecules is complex. The folding of proteins and the electrostatic screening of charges between them (Manning effect) plays a role. See (*Barrat and Joanny, 1997*) for a review. We can still make a rough estimate in the following way. We assume that macromolecules are mostly proteins. At physiological $\mathrm{pH} = 7.4$, three types of amino-acids carry a positive charge, Lysine (7%), Arginine (5.3%), Histidine (0.7%) while two others Aspartate (9.9%) and Glutamate (10.8%) carry a negative charge. Added to this, Histidine has a $\mathrm{pKa} = 6$ smaller than the pH so the ratio of [histidine neutral base]/[histidine charged acid] is $10^{\mathrm{pH-pKa}} = 25$. Hence the contribution of histidine may be neglected. The occurrence of the aforementioned amino acids in the formation of proteins is also known. The average length of proteins is roughly 400 amino acids. We subsequently obtain the average effective number of negative charges as,

$$z_c = 400(9.9 + 10.8 - 7 - 5.3)/100 = 25.$$

Such estimate needs to be refined and account for sugars and other macromolecules which carry more negative charges per chain but a interval from $z_c = 10$ to $z_c = 100$ charges is a plausible estimate.

The estimate of $\beta$ requires the knowledge of the physiological external concentration of ions $C_e = 150\mathrm{mM}$, $N_e = 140\mathrm{mM}$ and $K_e = 10\mathrm{mM}$ as well as conductances of sodium and potassium ions through the plasmic membrane. Here, again the situation is complicated since the dynamical opening of channels due to some change in the membrane potential (*Hodgkin and Huxley, 1952*) as well as the mechanical opening mediated by membrane stretching can play a role and affect these quantities. Nevertheless a rough estimate can be given (*Yi et al., 2003*)

$$g_N = 2 \times 10^{-6} \mathrm{C.V^{-1}.s^{-1}} \text{ and } g_K = 4.5\% \times 10^{-5} \mathrm{C.V^{-1}.s^{-1}}$$

Also the pump rate is estimated in *Luo and Rudy, 1991*,

$$j_p = 2.78 \times 10^{-12} \mathrm{mol.s^{-1}}.$$

This pump rate is in good agreement with the optimal pump rate predicted by the model,

$$j_p^{opt} = 3 \times 10^{-12} \mathrm{mol.s^{-1}}.$$

This leads to an estimate of

$$\beta = 0.1.$$

The density of macromolecules inside the cell is then found to be $\rho_c = 3 \times 10^6$ macromolecules per $\mu\text{m}^{-3}$ which is a correct order of magnitude (**Milo, 2013**). To further check the soundness of the above theory we can also compute the membrane potential and obtain $V_c = -73$ mV in good agreement with classical values .

## A.5 Osmotic compression of the cell

We now consider the case where, from the reference configuration, we impose an additional osmotic pressure in the external solution with Dextran polymers $\Pi_d = k_B T D_e$. We recall that according to formula (**Equation 22**), $\Pi = 0$ for small Dextran molecules while $\Pi = \Pi_d$ for big Dextran molecules.

We use (**Equation 31**) and (**Equation 32**) to compute the ensuing small displacement $u$. Assuming in good approximation that the osmotic pressure imposed by chloride ions is much larger ($10^5$ Pa) than the mechanical resistance of the cell cortex and the external matrix ($10^3$ Pa) $k_B T C_e \gg \tilde{k} R_c$ we find that,

$$u = -\frac{(z_c X_c)^{1/3}((\tilde{\alpha}-1)\Pi + \Pi_d)}{4\ 3^{2/3}\pi^{1/3}(1-\beta)^{7/6}C_e^{4/3}k_B T}.$$

Strinkingly, making the realistic simplifying assumptions that $K_d \gg G$ and $R_m \gg R_c$, leads to the same displacement of the cell membrane in the two situations of small and big Dextran:

$$u = -\frac{D_e(z_c X_c)^{1/3}}{4\ 3^{2/3}\pi^{1/3}(1-\beta)^{7/6}C_e^{4/3}},$$

showing that the two different osmotic loading are not distinguishable at that level. The main text relation (**Equation 1**) is obtained by assuming that the osmotic pressure of negatively charged ions is half the osmotic pressure of all ions.

However, the mechanical stress applied of the cell is completely different in both situations. For small Dextran, the mechanical stress confining the cell reads,

$$T^{\text{small}} = \mathbf{\Sigma}_m \mathbf{e}_r.\mathbf{e}_r|_{R_c} = \frac{2D_e G}{3C_e(1-\beta)}$$

while for big Dextran it reads,

$$T^{\text{big}} = \mathbf{\Sigma}_m \mathbf{e}_r.\mathbf{e}_r|_{R_c} = -\Pi_d + T^{\text{small}}.$$

Since $T^{\text{small}} \ll \Pi_d$ by at least one order of magnitude, the most important feature that changes between small and big Dextran is that $T^{\text{small}} > 0$ while $T^{\text{big}} < 0$. The physical picture behind this is that small Dextran compresses the cell without draining the water out of the matrix. Therefore, the cell behaves as a small inclusion which volume is reduced by the osmotic compression. In response, the matrix is elastically pulling back to balance the stress at the interface. In contrast, for big Dextran, the water is drained out of the matrix which therefore compresses the cell.

Notice that, like the membrane displacement, the variation of the membrane potential $v \overset{\text{def}}{=} v_c - V_c$ is the same in the two situations:

$$v = -\frac{k_B T}{q}\frac{D_e}{2\sqrt{1-\beta}C_e},$$

where we have made the same previous simplifying assumptions that $k_B T C_e \gg \tilde{k} R_c$, $K_d \gg G$ and $R_m \gg R_c$. Again such variation is negligibly small in our conditions where $D_e \ll C_e$ by several order of magnitudes. This further indicates that the biological response of the cell in response to a big Dextran compression has a mechanical rather than an electro-static origin.

## B Mechanical transmission of the stress through Matrigel

The purpose of this appendix is to verify that an extracellular matrix effectively transmits the mechanical stresses to the cells within it. To this end, we embedded soft polyacrylamide (PA) beads in a matrigel scaffold (see *Appendix 1—figure 7a*), before compressing the whole scaffold either with small or with big dextran. The polyacrylamide beads were fabricated as detailed in *Dolega et al., 2017*, they have a typical size of 20–50 μm, a bulk modulus modulus $K_{PA} \simeq 15$ kPa and are fluorescently labelled. We evaluated the volume reduction of the PA beads by imaging them before and after compression (*Appendix 1—figure 7b*). The bead volume was estimated by measuring the surface of its equatorial section, and assuming that the compression is isotropic. This assumption is valid only for beads located at the top of the dropled (z > 1 mm), where the interaction with the substrate is negligible. The bead volume change $\Delta V/V$ (*Appendix 1—figure 7c*) is measured under three experimental conditions:

- The culture medium is replaced by fresh medium with identical osmotic pressure (Control, n = 20 beads),
- The medium is replaced by fresh medium supplemented with small Dextran at $\Pi_d = 5$ kPa (Small, n = 20),
- The fresh medium is supplemented with big Dextran at $\Pi_d = 5$ kPa (Big, n = 18),

We observe that, when the MG is compressed by big dextran moelecules, the PA beads are also lose 25% of their volume, even though they are not directly in contact with the osmolytes (green). Such a volume loss is compatible with a mechanical pressure applied of the beads of few kPa (*Dolega et al., 2017*) and shows that the externally applied osmotic pressure results in a similar mechanical pressure applied on the beads through the drainage and compression of the MG meshwork. In contrast, if the dextran molecules are small enough to penetrate the MG and the PA beads, no mechanical stress is exerted on the inclusions (blue).

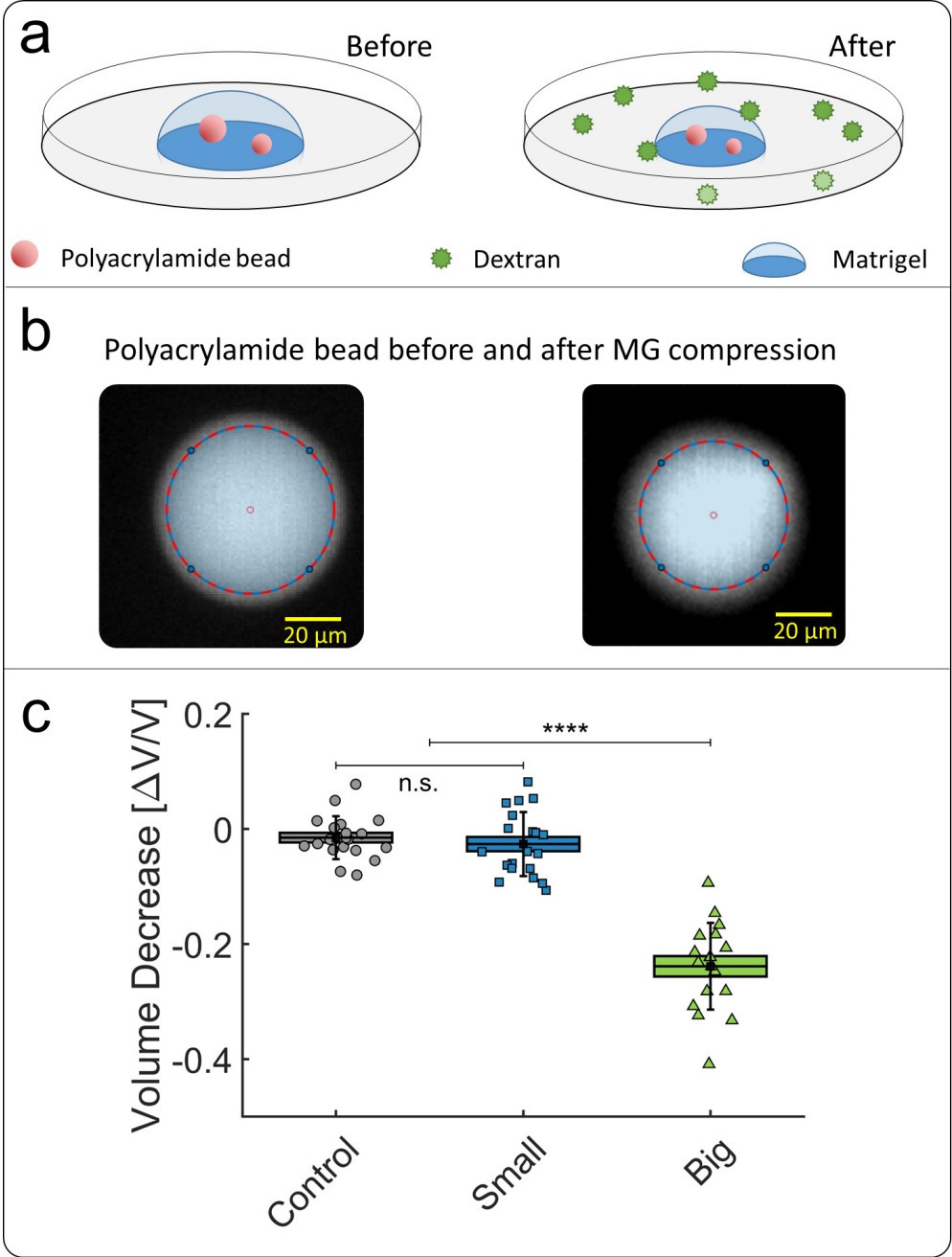

**Appendix 1—figure 2.** Compression of polyacrylamide beads embedded in Matrigel. (**a**) Schematic view of the experiment: compressible polyacrylamide beads (in red) are embedded in a matrigel drop and imaged before and after the addition of Dextran at $\Pi_d$ = 5 kPa. (**b**) Images of a fluorescent bead, respectively before and after compression. The bead volume is deduced from its equatorial section, assuming that the beads remain spherical after compression. (**c**) Volume decrease of polyacrylamide beads under compression occasioned by small (blue) and big (green) dextran.

## C Interstitial space
### C.1 Volume fraction of the intestitial space

To evaluate the volume fraction of the interstitial space in multi-cellular spheroids (MCS), we supplement the culture medium with sulforhodamine-B, a hydrophilic fluorophore that stains the extracellular space without penetrating the cells. From confocal sections of MCS (*Appendix 1—figure 3a*) we

determine the thickness of the thin layer between two adjacent cells. By fitting the intensity profile to a Gaussian distribution (*Appendix 1—figure 3b*), and taking into account that the instrumental function (resolution 270 nm) broadens the profile, we estimate the extracellular layer to 0.9 ± 0.1 µm (histogram in *Appendix 1—figure 3c*; N = 132). With an average cell diameter of 20 µm, we evaluate that the fraction of extracellular space is approximately $n_m = V_m/V_0 = 14 \pm 5\%$.

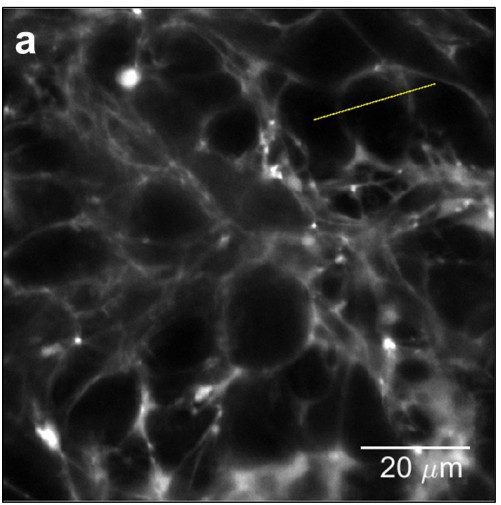

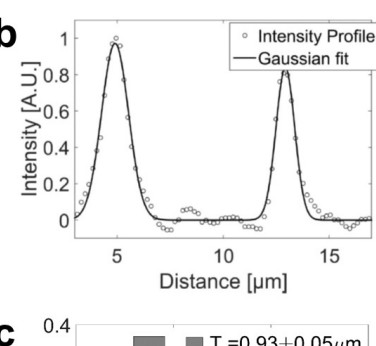

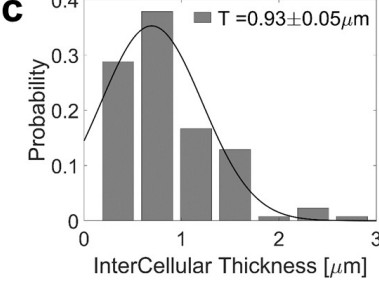

**Appendix 1—figure 3.** Volume fraction estimation. (**a**) Confocal section of a MCS, the extracellular space of which is filled with sulforhodamine-B. (**b**) Intensity profile across two extracellular layers. The width of intercellular space is computed by fitting the intensity to a gaussian profile. (**c**) Distribution of the intercellular layer thicknesses.

## C.2 Compression of ECM in the interstitial space of MCS

The evolution of the rheological properties of ECM filling the interstitial space of MCS is very difficult to evaluate, as the interstitial layer is extremely thin (see previous section). However, we can empirically define an exclusion-size (i.e. porosity), above which globular molecules do not penetrate the gel. To evaluate this exclusion-size, we dip the MCS in a solution containing fluorescent tracers with different radii. As shown in *Appendix 1—figure 4a*, tracers with $R_S$ = 4.4 nm and $R_S$ = 5.8 nm permeate the extracellular space of the MCS but not those larger than 14.8 nm. In order to quantify the relative amount of tracers inside the MCS, we compare the average fluorescence measured inside the MCS, $\langle I_{In} \rangle$ and in the surrounding solution $\langle I_{Out} \rangle$. *Appendix 1—figure 4b* reports the relative intensities $\langle I_{In} \rangle / \langle I_{Out} \rangle$, obtained respectively at an external osmotic pressure $\Pi_d = 0$ Pa and at $\Pi_d = 5$ kPa. In both cases, the fluorescence level lowers with large tracers. From the results presented in this section, we deduce that:

- The intercellular space is rich in fibronectin, a protein constitutive of the extracellular matrix;
- The intercellular space has a porosity comparable to that of matrigel gelified in vitro;
- The exclusion size of the intercellular space slightly decreases under compression, indicating a moderate compaction of the extracellular matrix.

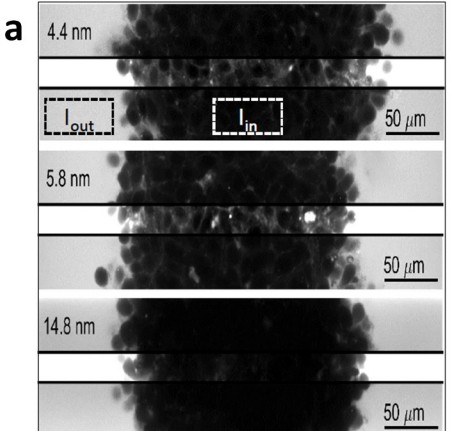
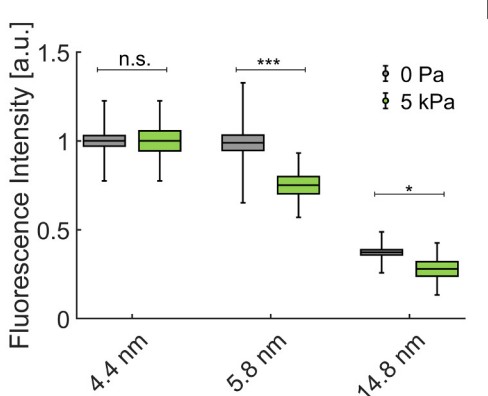

**Appendix 1—figure 4.** Exclusion size of the ECM in MCS. (**a**) Confocal sections of three MCS dipped in culture media supplemented with Dextran of increasing molecular weights. To avoid saturation of $\langle I_{Out} \rangle$, photomultiplier gain is kept low. This reduces the visibility of extracellular space inside the MCS. In the middle stripe of each image, the brightness is increased of the same amount to make the fluorescence of Dextran visible in the extracellular space. (**b**) Relative intensity $\langle I_{In} \rangle / \langle I_{Out} \rangle$ for different Dextran sizes, respectively, without pressure (gray boxes) and with 5 kPa (green boxes). Box sizes and error bars represent respectively the standard error of the mean and the standard deviation. Controls results (no pressure) are obtained from 58 MCS per condition. Experiments under pressure are averaged over 16 (for 4.4 nm), 14 (for 5.8 nm), and 13 (for 14.8 nm) MCS.

## D Cell volume

In order to estimate how cells react to an overall compression of the whole MCS, we measure the cell volume change within the aggregate. Cell contours are manually segmented from stack piles obtained with two-photon imaging. Manual segmentation is performed with Amira software. Cell volume is extracted before and after application of osmotic pressure for the same cell in order the measure its compression. Spherical cells are excluded of the analysis as they may undergo cell division and display rapid volume changes. Cells larger than 7000 μm³ are also discarded from the analysis as they may be two cells rather than one. The results are reported in *Appendix 1—figure 5* and show that cells appear to be compressed both by small (40 kPa) and big (15 kPa) dextran molecules. The pressures are chosen to match with the experiments presented in *Figure 3* of the main article. Notice that this method is much less accurate than the fluorescence exclusion' method used to determine the volume of individual cells. The results have to be taken as qualitative.

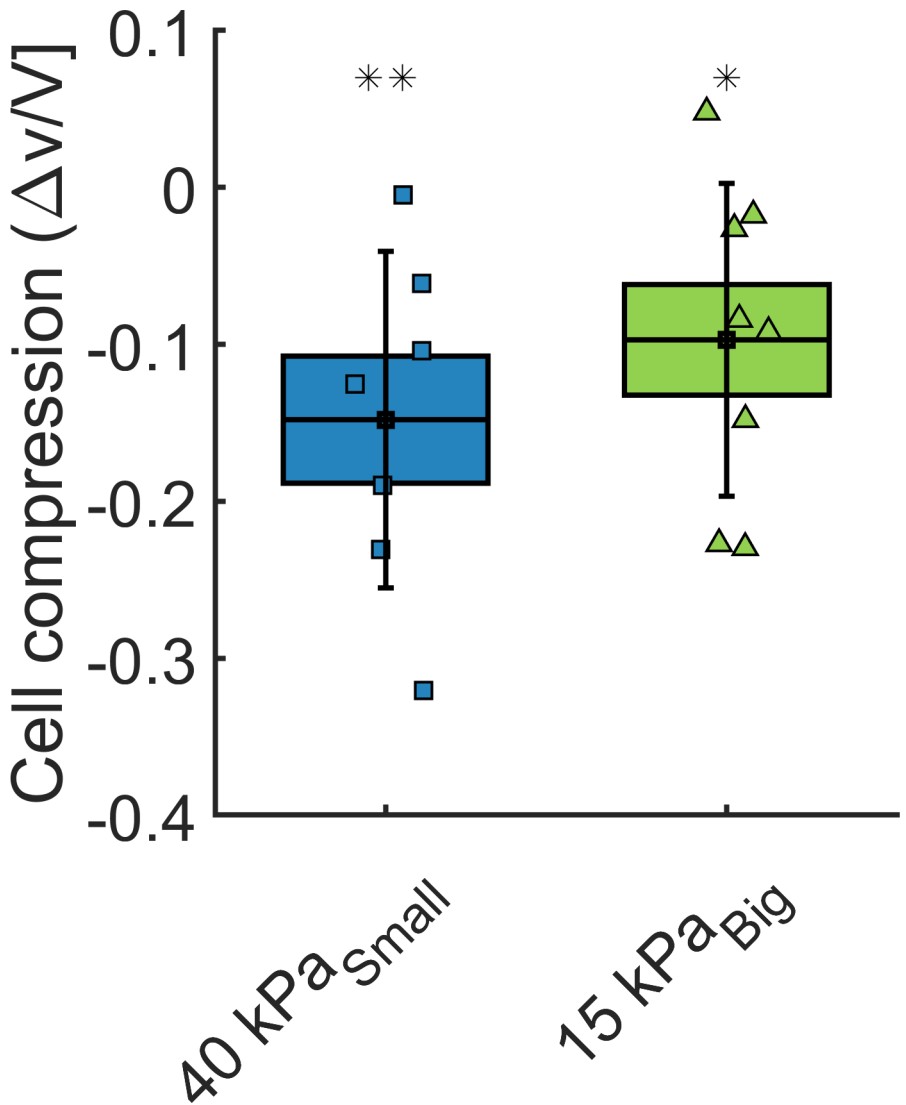

**Appendix 1—figure 5.** Cell volume change. Volume loss of cells inside the spheroids, under $\Pi_d^{Small} =$ 40 kPa and $\Pi_d^{Big}$ = 15 kPa, respectively.

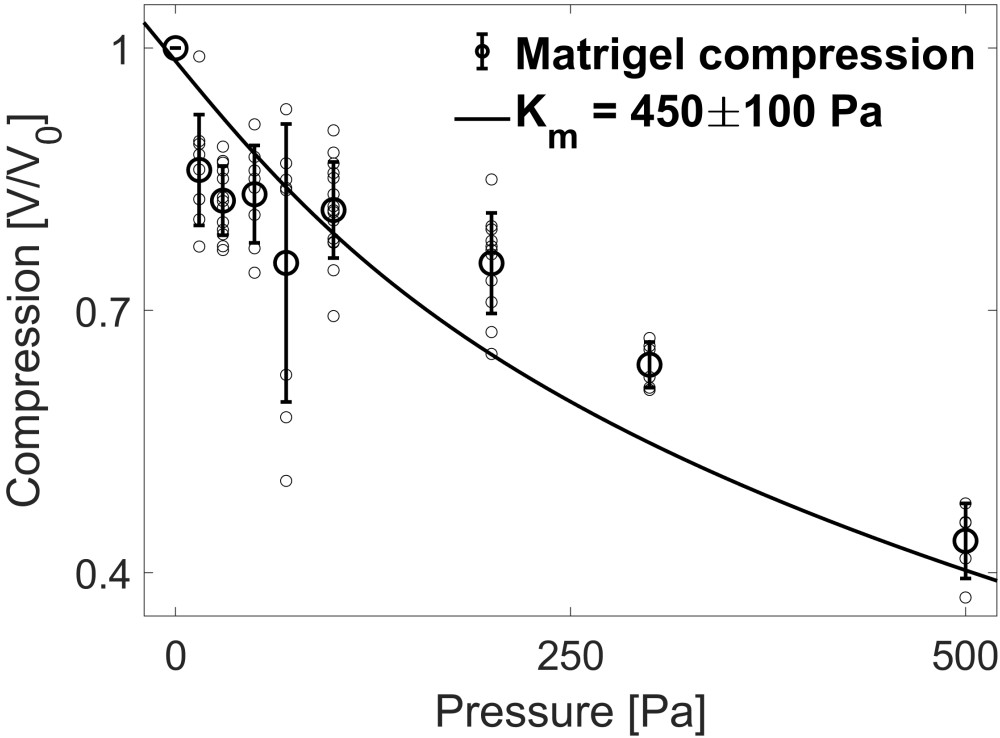

**Appendix 1—figure 6.** Matrigel bulk modulus. Compression of the Matrigel beads as a function of the osmotic pressure. Small circles correspond to individual measurements on different MG beads, large circles to the mean value at a given pressure and error bars to the standard deviation. Data are fitted to a hyperelastic model to determine the bulk modulus of matrigel (continuous line).

### E Matrigel bulk modulus

In this section, we estimate the bulk modulus of the extracellular matrix. As interstitial ECM is difficult to characterize in-situ, we use matrigel (MG) beads to roughly estimate the rheological properties of ECM. Consistently with native ECM, large Dextran molecules were also excluded from microbeads made of MG suggesting an equivalent effective permeability (*Dolega et al., 2021*). To determine the bulk modulus of MG beads, we follow their compression at different dextran concentration. To facilitate the measurement, the beads are doped with fluorescent nanoparticles. In *Appendix 1—figure 6*, we display the volume decrease $V/V_0$ of MG beads as a function of the osmotic stresses, between 15 and 500 Pa ($V_0$ being the bead volume before compression). The continuous line represents the best fit to a Mooney-Rivlin model, the derivative of which represent the bulk modulus $K_m$(*Rivlin and Saunders, 1951*). For small deformations, the best fit is obtained for a bulk modulus $K_m = 450 \pm 100$ Pa.

### F Dynamic light scattering and motile activity inside MCS

The scope of this section is to illustrate how to determine cell motility inside an opaque multicellular aggregate. Previous works already indicate that pressure affects cell motility in multicellular spheroids, but the observations are limited to either the superficial layer (*Alessandri et al., 2013*) or to the long-term centripetal motion (*Delarue et al., 2013*). Recently, we developed a method to measure the cell velocity in the deep layers of MCS without using confocal microscopy, which is limited in terms of sample thickness and observation time. In our setup (*Brunel et al., 2017*; *Brunel et al., 2020*; *Appendix 1—figure 7a*), the MCS is observed by phase contrast (*Appendix 1—figure 7c*) and is simultaneously illuminated with an infrared laser (850 nm). The light scattered by the MCS in the forward direction produces an interference pattern, which is collected by a camera (*Appendix 1—figure 7b*). From the of temporal fluctuations of this pattern (signal shown in *Appendix 1—*

figure 7d and its autocorrelation function in *Appendix 1—figure 7e*), one computes the average velocity of cells, moving inside the MCS. It has to be noticed that this technique, an evolution of the Dynamic Light Scattering, provides information on the 3D motility, and not only on the 2D motion as previously measured by *Alessandri et al., 2013* at the surface of MCS. With this method, we measure the average speed in the three cases of interest: without pressure, when a pressure is selectively applied on the cells, but not on the ECM (small Dextran), and when the pressure is applied to the whole MCS (big Dextran). The results are shown in *Appendix 1—figure 7f and g*: whereas the average speed is comparable in the first two cases (10 ± 1 µm/hr; magenta and cyan), it is reduced by a factor of two when the compression is exerted on the entire MCS (4.8 ± 0.5 µm/hr, blue).

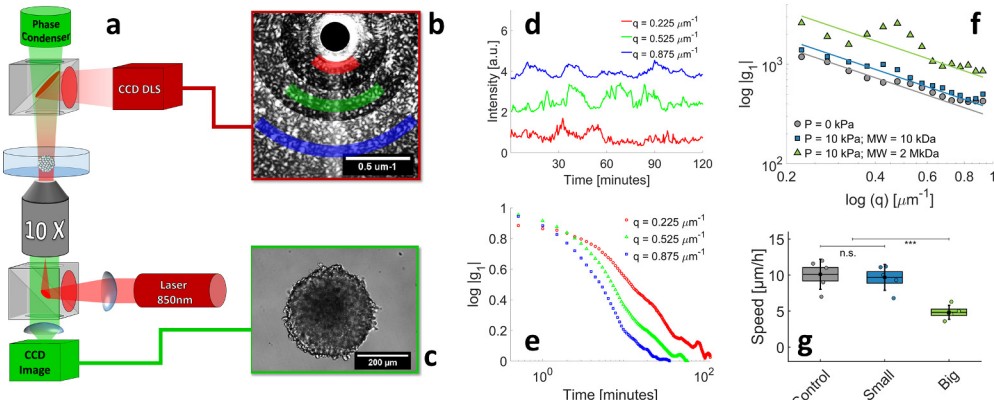

**Appendix 1—figure 7.** Motile activity measured by Dynamic Light Scattering (DLS) under different pressure conditions. (**a**) The experimental setup combines two counter-propagating optical pathways, with two different wavelengths. This allows us to observe the MCS simultaneously by DLS ($\lambda_{DLS}$ = 850 nm, dark red in the sketch) and by phase contrast ($\lambda_{phase}$ = 530 nm, green). The DLS signal and phase contrast image are illustrated respectively in panels (**b**) and (**c**). DLS signals are acquired at different scattering vectors q and averaged over rings of equal q=|q| (colored sectors in panel (**b**)). (**d**) Time evolution of diffraction intensity at three different q; colors correspond to that of sectors in panel (**b**). (**e**) Intensity-Intensity autocorrelation functions for the three different scattering vectors. In the single-scattering regime, the intensity signal decorrelates in a typical timescale $\tau = 1/qv^0$, where $v^0$ is the mean cell velocity inside the MCS. (**f–g**) respectively the decorrelation time as a function of q and the resulting mean cell velocity, measured in three different conditions: with no pressure, $v^0$ = 10 ± 1 µm/hr (magenta), with 5 kPa exerted by small Dextran, $v^0$ = 9.8 ± 0.8 µm/hr (cyan), and with $\Pi_e$ = 5 kPa exerted by large Dextran, $v^0$ = 4.8 ± 0.5 µm/hr (blue). Box sizes correspond to the standard error of the mean (N = 5 MCS) and the error bars to the standard deviation.

## G Cytoskeleton and compressibility

Aggregate compressibility depends primarily on three elements: the compressibility of the cells, that of the extracellular matrix, and the volumetric ratio between the two. In this section, we want to evaluate how the rheology of the cytoskeleton contributes to the apparent compressibility of MCS. To do so, we prepare six 96-wells plates of identical spheroids. To avoid the formation of a necrotic core, the MCS initial radius does not exceed 200 µm. Subsequently, five drugs and beg dextran molecules (2 MDa) are added to the different plates:

* Y-27632 ROCK inhibitor, to reduce cell contractility,
* Blebbistatin, to inhibit acto-myosin activity,
* Cytochlasin D, to inhibit actin polymerization,
* Nocodazole, to promote microtubule depolymerization,
* Paclitaxel, to impede microtubule depolymerization.

As a control, a subset of spheroids are exposed either to the drug alone, without dextran in solution or to Dimethyl Sulfoxide (DMSO). The MCS volumes are obtained by measuring the surface of their equatorial planes and considering them as a perfectly spherical object. The volume is measured

before adding the drug, then 45 min after exposure to the drug alone (gray in *Appendix 1—figure 8*) or to the drug supplemented with dextran (green in *Appendix 1—figure 8*) and normalized to the initial volume of each MCS.

We observe that drugs modify the MCS volume in different manners (see the figure below) as compared to the control (DMSO). This is in agreement with the fact such pharmacological perturbations are kwown to impact the single-cell volume in different maners (*Stewart et al., 2011*). When dextran is added to the solution, the spheroids get compressed from their initial state (with the drug); such final compression (Dextran+Drug) is comparable to that obtained with dextran alone but the amout of compression with respect to the intial state varies depending on the drug.

This result is compatible with our idea the 5 kPa Dextran compression reduces almost to the maximum the inter-cellular space and that cells are then almost fully connective in the final state. Thus, depeding on the amount of compression that the drug first creates, the ensuing compression with Dextran will change depending on the available inter-cellular space that remains. For instance, in the presence of cytochalasin, the extra-cellular space is already largely reduced compared to DMSO so when the osmotic compression follows, their is hardly no inter-cellular space which can still be compressed. This seems to be an additional indication that the MCS compressibility in response to a gentle osmotic pressure is more related to the rheology of the extracellular space than to the internal organization and contractility of the cytoskeleton as we argue in *Dolega et al., 2021*.

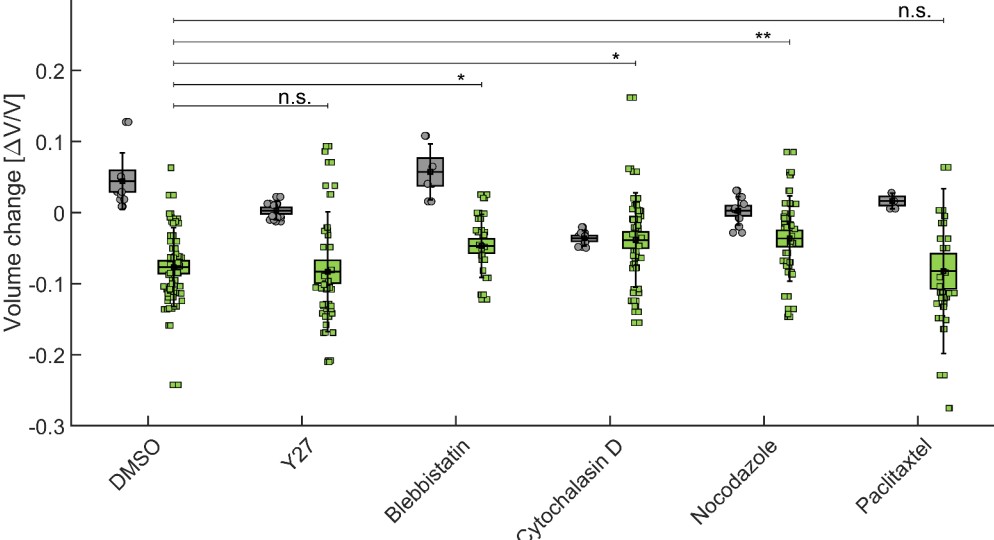

**Appendix 1—figure 8.** Influence of cytoskeleton structure and contractility on MCS compressibility. In gray, the relative change of the MCS volume after injection of drug (or DMSO alone in control experiments). In green, the vovlume change after addition of drug+dextran at 5 kPa. BoxPlot represent the standard error of the mean, error bars the standard deviation, and circles/squares the volume change of individual spheroids.

