## [Decision Letter]

**Acceptance summary:**

This paper provides an elegant approach using crowding agents to study how distinct types of external mechanical perturbations influence the behavior of multicellular aggregates. The central finding is that cell proliferation and motility are significantly affected by solid stresses mediated by the extracellular matrix, whereas osmotic stresses of a similar magnitude have little effect. This finding highlights a mechano-sensory function of the extracellular matrix to regulate cell behavior in 3D environments.

**Decision letter after peer review:**

Thank you for submitting your article "Extra-cellular Matrix in cell aggregates is a proxy to mechanically control cell proliferation and motility" for consideration by *eLife*. Your article has been reviewed by three peer reviewers, and the evaluation has been overseen by a Reviewing Editor and Aleksandra Walczak as the Senior Editor. The reviewers have opted to remain anonymous.

The reviewers have discussed the reviews with one another and the Reviewing Editor has drafted this decision to help you prepare a revised submission.

Summary:

Dolega et al. investigate how proliferation and motility of cells inside multicellular aggregates are affected by different types of compression such as an osmotic pressure or a solid stress mediated by an extracellular matrix. They develop a selective compression method using crowding agents like dextran of different sizes: by using dextran that is small enough to penetrate the matrix, they impose an osmotic stress on the cells, while larger non-penetrating dextran instead results in a solid stress exerted by the matrix. The central find of this work is that cell proliferation and motility are significantly affects by solid stresses mediated by the extracellular matrix, whereas osmotic stresses of a similar magnitude have little effect.

Essential revisions:

The reviewers indicate that this is an interesting study and express enthusiasm about the ideas proposed in this study. However, several concerns are raised about the experimental evidence to support the main claims. Among other things, reviewer #2 points out that a better characterization of the ECM and the compressive deformations that are induced by the selective compression method is needed, as well as the induced compression of the cells. reviewer #3, raises similar issues, and suggest approaches to determine the stresses induced by the various types of dextran. While reviewer #1 seems quite positive overall, they also make the point that the precise mechanical role of the ECM in the observed behavior is not pinned down, and suggest that experiments that would clarify this would be a great addition. Together, the reports indicate the main claims of the manuscript would need to be more strongly supported by showing experimentally that the ECM is really produced by the cells and that there is ECM between the cells in the multicellular spheroids. In addition, the reviewers point out that direct evidence for induced stresses or deformations in the ECM is currently lacking. Finally, several comments from all reviewers raise issues about the statistical characterization of the data and possibly missing controls.

Reviewer #1:

This is an excellent study, which clearly demonstrates that the osmotic compression of the extracellular matrix (ECM) with large osmolytes reduces the growth rate of cells and their migration speed. In contrast, the same concentration of small osmolytes, which can infiltrate inside the ECM, results in a small tension of the ECM, while the growth rate of cells and their migration speed remain unaffected.

As the authors mention in the last paragraph of the Discussion, it remains unclear whether the observed behavior is due to the transmitted stress from the compressed ECM to the cells or due to the increased density of the compressed ECM. The authors mention that this could be resolved in the future by performing additional control experiments by varying the ECM density without any imposed osmotic pressure. Can the authors comment on how easy or hard it is to control the ECM density either in the matrigel or in the multicellular spheroids? If such experiments are straightforward, they would be a great addition to this paper.

Reviewer #2:

The manuscript provides an interesting work studying the effect of ECM transmitted compression on cancer cell proliferation and mobility. The main claim is that the ECM signals if cells are under global or local compression, thus resulting in different cell behavior. Although I really like the ideas proposed, I fear that the current data does not sufficiently support the claims. Overall the interpretation of the experimental data is largely exceeding what was actually shown. A simple example is that the authors constantly talk about ECM between cells in the MCS, however they never demonstrate that this ECM is indeed there. Unfortunately, there are a number of similar problems that largely reduce the quality of the work, making it currently unsuitable for publication in *eLife*.

1) My biggest concern is that the main claim of the work is not supported by direct measurements. It is claimed that the reduction in proliferation and motility is because of the compression transmitted by the ECM. However, the ECM and its deformation is not measured in the MCS experiments. It is simply assumed that it is there, and that it has the same properties as a individually polymerized Matrigel (MG). It is well established that the ECM organization by cells is different from repolymerized gels. The authors argue with previous studies, however, as this is the main point of the paper it is important to directly demonstrate the presence and the effect of the force transmission of the ECM in situ.

A possible experiment would be to prevent the ECM from forming in the spheroids, or to degrade it previous to the experiment. Alternatively the authors could try to use an inert hydrogel.

The very simple alternative to the ECM transmitted forces is that the cell directly push on each other, hence the ECM becomes irrelevant, as discussed by the authors. The authors try to convince the reader by the experiments of single cells in MG (Figure 5). However a simple explanation could be that due to pressure the density of the MG is beyond the point that motility is possible, hence the cells start clustering, and a local decrease of nutrients may explain the reduced proliferation.

Unfortunately, the data presented does not sufficiently demonstrate the claims.

2) It is quite irritating to be confronted with different pressures, and subsequent comparisons. Why not show the effect of 0,5,10 and 15 kPa pressure in all plots? Also the authors should provide statistical tests in all presented plots.

3) Simple controls are missing. What is the speed and proliferation of these cells in a 2D surface without dextran, and with the different dextran types and concentrations (to exclude effects of Dextran itself)?

4) The argument that commercially ECM is the same as cancer cells generate is not sufficiently backed regarding the importance of this point for the paper. You have to check if this is the case for the used cells, and if this is also happening in the spheroids.

5) A main claim at the end of the subsection “Selective-compression method” is that the single cell compression can be used as proxy for the individual cell compression in the spheroids. I fear given the importance of this point for the main message it is necessary to measure the compression of the individual cells in the spheroid. This should be possible looking at the data presented in Figure 3 (i.e. using cell profiler).

6) I am puzzled by the strong signal of KI67 in MCS without pressure. This suggests that almost all cells in the outer region are dividing, but the division time is 36 h. Since the mitosis needs about 1h, only 3% of the cells should be in mitosis. Here the thresholding is quite important and can lead to a wrong impression. Typically there should be no expression of KI67 during the time in between division (hence in more than 30h).

7) I think an important check for the overall hypothesis can be done with the cell confiner. Polymerizing single cells in Matrigel without the dextran, and then confining it with the instrument should give the same phenotype as the big dextran. I suggest to do this experiment.

Reviewer #3:

In this manuscript, Dolega and colleagues examine the impact of osmotic pressure versus solid stress (or "global compression" as the authors describe) on proliferation and motility in cell aggregates cultured in 3D gels. They develop a clever method of using dextrans that are small, so they can penetrate the gel, or are large, so that they are excluded from the gel, to exert increased osmotic pressure or solid stress, respectively. They find that solid stress impacts cell proliferation and migration within clusters, whereas increased osmotic pressure has a limited impact.

The question of how solid stress impacts cells is an important question, which is particularly relevant to the context of tumor growth. Approaches to test the impact of uniform solid stress on cells are limited. It is often assumed that modulating osmotic pressure can simulate solid stress. However, the authors show here that this assumption is incorrect. Thus, the approach the authors develop to modulating solid stress, and the finding that solid stress has a different impact than increased osmotic pressure, is important and a nice advance for the field, which is in principle suitable for *eLife*. This approach could serve as a powerful tool for the community. However, the method and biological results need to be better characterized in order for this manuscript to be suitable for publication in *eLife*. Additional no mechanistic insights are provided; some mechanistic insight would make the manuscript more impactful.

1) Characterization of method: How do the authors determine the solid stress for the large dextran used? Is it just assumed that whatever osmotic pressure is applied will translate to that amount of solid stress? While this might be the case, these gels are complex, some direct experimental measurement of this would build more confidence (e.g. by using a rheometer to measure negative normal stress of the gels or polyacrlymide hydrogel "reporter" beads). In addition, showing how solid stress varies as a function of big dextran density, and demonstrating the applicability of this idea in another common 3D culture gel system (e.g. collagen, PEG, agarose, or alginate) would facilitate the broader adoption and use of this technique.

2) Some of the experiments need more detail and statistical analysis:

Figure 2B, C and D – statistical analysis are needed.

Figure 4A: pattern and/or levels of proliferation should be quantified. Single representative images are not sufficient: quantitative data supported by statistically significant differences are required to make conclusions.

Figure 4B: are differences statistically significant? Statistical analyses should be included. Also, representative time series of images could be included.

Figure 4C: timelapse studies should be conducted, and timelapse images included, to better display the motion that is described.

Figure 5: statistical analysis should be included for D and F.

Figures 4 and 5: multiple levels of solid stress would be ideal to show the trend.

Figure 5: do cells extend beyond the periphery of the Matrigel beads?

3) Mechanistic insights – there are really no mechanistic insights provided as to how increased solid stress might lead to some of the behaviors observed. This is unusual for *eLife* papers. Even a screen of implicated small molecule inhibitors (e.g. actin, myosin, Rho,.…) could be valuable in generating some general idea into mechanism.

---

## [Author Response]

Reviewer #1:This is an excellent study, which clearly demonstrates that the osmotic compression of the extracellular matrix (ECM) with large osmolytes reduces the growth rate of cells and their migration speed. In contrast, the same concentration of small osmolytes, which can infiltrate inside the ECM, results in a small tension of the ECM, while the growth rate of cells and their migration speed remain unaffected.As the authors mention in the last paragraph of the Discussion, it remains unclear whether the observed behavior is due to the transmitted stress from the compressed ECM to the cells or due to the increased density of the compressed ECM. The authors mention that this could be resolved in the future by performing additional control experiments by varying the ECM density without any imposed osmotic pressure. Can the authors comment on how easy or hard it is to control the ECM density either in the matrigel or in the multicellular spheroids? If such experiments are straightforward, they would be a great addition to this paper.

We agree with the reviewer that this point is crucial. A partial answer to the question comes from the experiments added in Figure 5G of the revised version. We measured the proliferation of individual cells seeded in matrigel of increasing densities, but with no pressure. In the range 2-to-8 g/l, the MG density has a non-measurable effect. However, this measurement is not fully satisfying, as we should be able to vary the bulk modulus, the shear modulus and the biochemistry of the matrix independently. Unfortunately, this is long-term project, as the chemistry of the extracellular matrix is poorly known and its rheology depends in a non-trivial manner on the chemical composition. For instance, we have been surprised to measure that the Young modulus of MG decreases at high concentration (Atomic force microscopy experiments). This is probably because the thinning agent (Dulbecco medium), modifies the pH and the ionic strength of the solution and, thus, the MG polymerization.

Reviewer #2:The manuscript provides an interesting work studying the effect of ECM transmitted compression on cancer cell proliferation and mobility. The main claim is that the ECM signals if cells are under global or local compression, thus resulting in different cell behavior. Although I really like the ideas proposed, I fear that the current data does not sufficiently support the claims. Overall the interpretation of the experimental data is largely exceeding what was actually shown. A simple example is that the authors constantly talk about ECM between cells in the MCS, however they never demonstrate that this ECM is indeed there. Unfortunately, there are a number of similar problems that largely reduce the quality of the work, making it currently unsuitable for publication in eLife.1) My biggest concern is that the main claim of the work is not supported by direct measurements. It is claimed that the reduction in proliferation and motility is because of the compression transmitted by the ECM.

We agree that this is the crucial point of our article. Thus, we will try to convince the reviewer by giving a point to point answer.

However, the ECM and its deformation is not measured in the MCS experiments. It is simply assumed that it is there, and that it has the same properties as a individually polymerized Matrigel (MG).A possible experiment would be to prevent the ECM from forming in the spheroids, or to degrade it previous to the experiment.

We can provide two evidences that the ECM is present in multicellular spheroids made of CT26 cells: first, CT26 spheroids disassemble when the ECM is degraded using collagenase. Collagen seems to be essential to maintain spheroid cohesion. Second, immunostaining indicates the presence of fibronectin in the interstitial space. An image showing the localization of fibronectin was added in the revised manuscript (Figure 1A).

It is well established that the ECM organization by cells is different from repolymerized gels. The authors argue with previous studies, however, as this is the main point of the paper it is important to directly demonstrate the presence and the effect of the force transmission of the ECM in situ.

In the manuscript, we do not need to assume that MG has the same properties as ECM. We show that it is not even necessary to use the native ECM, but a non-specific and commonly used matrix (matrigel) is sufficient to reproduce the phenotype observed in MCS under compression (results on cells isolated in matrigel).

We also added a section in Appendix, with the confocal images used to estimate the porosity of the interstitial space. These results indicate that the ECM pore-size has the same magnitude as that of matrigel, i.e. ~10 nm.

Alternatively the authors could try to use an inert hydrogel.

An inert hydrogel is even further from ECM as compared to MG.

The very simple alternative to the ECM transmitted forces is that the cell directly push on each other, hence the ECM becomes irrelevant, as discussed by the authors. The authors try to convince the reader by the experiments of single cells in MG (Figure 5.). However a simple explanation could be that due to pressure the density of the MG is beyond the point that motility is possible, hence the cells start clustering, and a local decrease of nutrients may explain the reduced proliferation.

We disagree with this interpretation. The proliferation of individual cells in MG is followed for few days. We observe a clear effect of pressure already at day 3, when the cells have divided twice. At this timepoint, the spherical structures are made of at most 4 cells. Thus, they have full access to oxygen and nutrients present in the surrounding medium. We also now show on Figure 5G that increasing the density of MG has almost no effect on the proliferation rate.

2) It is quite irritating to be confronted with different pressures, and subsequent comparisons. Why not show the effect of 0,5,10 and 15 kPa pressure in all plots?

All the biological experiments (Figure 4 and 5) have been done at 5 kPa.

This choice was motivated by our previously published experiments (Montel et al., 2012; Brunel et al., 2020) and reported in Author response image 1. Such experiments showed that 5 kPa is the value at which the effect of osmotic pressure on proliferation saturates. At lower pressure proliferation and motility inhibition is less clear. Larger pressures do not exacerbate the effect, but approach the regime of high osmotic pressures, where other cell responses are involved.

Only in Figures 2 and 3, we present experiments at 15 kPa and 70 kPa. This has been done to validate the method, as 5 kPa did not give a measurable effect on the compression of individual cells or on the compression/swelling of the interstitial space.

We added an inset to Figure 3A, to show the volume loss of submitted to Π = 15 kPa (big dextran) and Π = 70 kPa (small dextran), to allow the reader to compare with the methodological experiments cited above.

To better explain these points, we have modified the text in the revised version.

Also the authors should provide statistical tests in all presented plots.

We apologize for this omission. Statistical tests have been added to all plots.

3) Simple controls are missing. What is the speed and proliferation of these cells in a 2D surface without dextran, and with the different dextran types and concentrations (to exclude effects of Dextran itself)?

We have measured that the proliferation and the velocity of cells seeded on a Petri Dish are not affected by the presence of dextran at 5 kPa. Results have been added in the new Figure 4F and G.

4) The argument that commercially ECM is the same as cancer cells generate is not sufficiently backed regarding the importance of this point for the paper. You have to check if this is the case for the used cells, and if this is also happening in the spheroids.

As explained above, we disagree with this comment. The ECM varies from cell type to cell type even in vivo. Our argument is not that MG resembles to the native ECM of CT26, but that a generic matrix allows us to reproduce the phenotype observed in MCS.

5) A main claim at the end of the subsection “Selective-compression method” is that the single cell compression can be used as proxy for the individual cell compression in the spheroids. I fear given the importance of this point for the main message it is necessary to measure the compression of the individual cells in the spheroid. This should be possible looking at the data presented in Figure 3 (i.e. using cell profiler).

We thank the reviewer for this suggestion. Cell profiler did not work well in our case, thus cells have been segmented manually from images with labelled intercellular space. Unfortunately, this method is very inaccurate, and would require the generation of fluorescent cell lines with membrane markers rather than staining the intercellular space to deliminate the cell contours. Direct measurement of volume of cells with non-regular shapes is a tricky problem even for single cells in 2D, so for cell aggregate it represents a real challenge. Nevertheless, our rough approach shows that cells are qualitatively compressed in both cases, with big and small dextran.

The results have been added the Appendix in the revised version.

6) I am puzzled by the strong signal of KI67 in MCS without pressure. This suggests that almost all cells in the outer region are dividing, but the division time is 36 h. Since the mitosis needs about 1h, only 3% of the cells should be in mitosis. Here the thresholding is quite important and can lead to a wrong impression. Typically there should be no expression of KI67 during the time in between division (hence in more than 30h).

We respectfully disagree. According to the seminal articles by Gerdes et al., 1984, Ki67 is present through the whole cycle (not only in itosis), but absent in G0. We added to the revised version this reference, together with a short description of Ki67. The reviewer probably refers to Edu, which is incorporated during the S-phase.

7) I think an important check for the overall hypothesis can be done with the cell confiner. Polymerizing single cells in Matrigel without the dextran, and then confining it with the instrument should give the same phenotype as the big dextran. I suggest to do this experiment.

We disagree because a “hydrostatic” and “uniaxial” loading are different. A “hydrostatic” stress field correspond to a homothetic loading in all spatial directions as investigated in the article. Instead, the reviewer suggests to apply a uniaxial compression, which will deform the cell by activating a shear mode rather than compress it. Whereas this is certainly a very interesting experiment (that is currently under investigation in M. Piel’s team (Institut Curie)), it would neither confirm nor refute the experiments performed under Dextran pressure.

Instead, we have embedded polyacrylamide beads in MG droplets. Polyacrylamide (PA) beads are compressible with a bulk modulus K_PA_~15 kPa, which is considerably higher than that of MG K_MG_<1 kPa. When big dextran molecules surround the MG droplet, the embedded PA beads get compressed, even though they never get in contact with the osmolyte. This experiment, which shows that the matrix transmits the stress to the inclusions, has been added to the appendix B of the revised version.

Reviewer #3:In this manuscript, Dolega and colleagues examine the impact of osmotic pressure versus solid stress (or "global compression" as the authors describe) on proliferation and motility in cell aggregates cultured in 3D gels. They develop a clever method of using dextrans that are small, so they can penetrate the gel, or are large, so that they are excluded from the gel, to exert increased osmotic pressure or solid stress, respectively. They find that solid stress impacts cell proliferation and migration within clusters, whereas increased osmotic pressure has a limited impact.The question of how solid stress impacts cells is an important question, which is particularly relevant to the context of tumor growth. Approaches to test the impact of uniform solid stress on cells are limited. It is often assumed that modulating osmotic pressure can simulate solid stress. However, the authors show here that this assumption is incorrect. Thus, the approach the authors develop to modulating solid stress, and the finding that solid stress has a different impact than increased osmotic pressure, is important and a nice advance for the field, which is in principle suitable for eLife. This approach could serve as a powerful tool for the community. However, the method and biological results need to be better characterized in order for this manuscript to be suitable for publication in eLife. Additional no mechanistic insights are provided; some mechanistic insight would make the manuscript more impactful.1) Characterization of method: How do the authors determine the solid stress for the large dextran used? Is it just assumed that whatever osmotic pressure is applied will translate to that amount of solid stress? While this might be the case, these gels are complex, some direct experimental measurement of this would build more confidence (e.g. by using a rheometer to measure negative normal stress of the gels or polyacrlymide hydrogel "reporter" beads). In addition, showing how solid stress varies as a function of big dextran density, and demonstrating the applicability of this idea in another common 3D culture gel system (e.g. collagen, PEG, agarose, or alginate) would facilitate the broader adoption and use of this technique.

Theory: We first would like to conceptually clarify the classical situation of the osmotic compression of a passive poroelastic material. To do so, we first explain theoretically why an osmotic shock with a osmolite that cannot permeate the mesh of the poro-elastic material leads to a mechanical strain (and hence a solid stress) in the material.

In the revised version we experimentally show that this idea is correct on the simple example of Matrigel that is used in experiments of the paper. The experimental part is added as an appendix of the new manuscript.

Poroelasticity is a theory describing the flow of a liquid in a solid deformable porous medium usually called the skeleton. The premise of the theory dates back to Gibbs and was then pioneered by M.A. Biot [Journal of applied physics, vol 12, pp.155-164, 1941]. A justification of the theory from a homogenization perspective as well as its thermodynamic foundations can be found in the textbook of O. Coussy [Coussy, 2004]. While, for convenience of the reviewer, we will here only recall the “small displacements” theory, the same derivation can be followed in the geometrically non-linear setting although it becomes much more technical.

We consider a fluid-infiltrated solid medium which occupies the domain 𝛺, immersed in the fluid. We denote the spatial coordinate *r* ∈𝛺 and the time *t.* Inside this domain, in the absence of inertial effects and body forces, balance of momentum reads,∇. σ¯=, (1)where σ̿ (𝑟,𝑡) is the total –solid plus fluid- stress. We use the classical differential notation where 𝛻. is the divergence operator. The effective rheology of a poro-elastic material can be formulated in the following way. First we relate the stress with the strain and the hydrostatic pore pressure 𝑝(𝑟, 𝑡) (with respect to the pressure in the fluid bath) asσ̿=2Gε̿+(K−2G3)tr(ε̿)I̿−αpI=,(2)

Where the strain is conventionally defined as 𝜀̿ (𝑟, 𝑡) = (𝛻𝑢 + 𝛻𝑢^𝑇^)/2 with 𝑢(𝑟, 𝑡) the displacement field of the skeleton from the rest state. 𝐼̿ denotes the identity matrix. Relation (2) generalizes the elastic Hooke’s law for a poro-elastic medium.

This description introduces some rheological coefficients:

– 𝐾 is the so-called drained bulk modulus corresponding to deformations of the material at fixed 𝑝 = 0 when the fluid can flow in and out of the skeleton.

– 𝐺 is the shear modulus of the skeleton and is unaffected by the presence of the permeating fluid since fluid cannot resist shear.

– 𝛼 is the Biot-Willis coefficient which measures the amount of pore pressure contributing to total stress. For biological materials this parameter is generally very close to one because they are composed of highly compressible material permeated by an almost incompressible fluid. We shall assume that 𝛼 = 1.

In steady state, the pore pressure equilibrates with the applied osmotic pressure 𝜋_𝑑_ such that 𝑝 = −𝜋_𝑑_ (see [Dolega et al., 2020] for a justification ). In the case of a spherical ball, thanks to the symmetries, the solution of the ensuing mechanical problem is

𝜎 (𝑟,𝑡) = 0. This means that the elastic stress in the skeleton is compensated by the pore pressure term 𝑝. Hence, using the constitutive relation (2) , we obtain,−πdI̿=2G∈̿+(K−2G3)tr(ε̿)I̿.

Taking the trace of this equation, we obtain the volumetric strain in the skeleton following the application of 𝜋_𝑑_ istr(∈̿)=∇VV=−πdK.

This shows that by applying an osmotic pressure to a poro-elastic ball, we induce a solid stress in its skeleton which is exactly −𝜋_𝑑_. This leads to a constant volumetric strain −𝜋_𝑑_/𝐾 inside the ball. This is why the drained bulk modulus 𝐾 of the skeleton (as referred to in the mechanics community) is also often also called the osmotic modulus.

Experiments: We experimentally demonstrate that this theoretical idea is applicable to Matrigel in the main paper. We exploited the reviewer's idea of embedding deformable beads in a matrigel scaffold, to experimentally prove that the extracellular matrix is capable of transduce the osmotic pressure into solid stress. These experiments were added in Appendix B of the revised manuscript.

2) Some of the experiments need more detail and statistical analysis:

We thank the reviewer for accurately pointing all those omissions.

Figure 2B, C and D – statistical analysis are needed.

Added in the revised version.

Figure 4A: pattern and/or levels of proliferation should be quantified. Single representative images are not sufficient: quantitative data supported by statistically significant differences are required to make conclusions.

A panel was added to Figure 4 to quantify the density of proliferating cells (Ki67 positive) as a function of the radial position, in the three conditions. The text and the legend were modified accordingly.

Figure 4B: are differences statistically significant? Statistical analyses should be included. Also, representative time series of images could be included.

Statistical analysis and time series of images has been added to Figure 4, in the revised version.

Figure 4C: timelapse studies should be conducted, and timelapse images included, to better display the motion that is described.

There are no images, as the velocity has been measured by Dynamic Light Scattering. In fact, cells cannot be imaged inside the spheroids, as light scattering makes the spheroid opaque beyond the two first cell layers. Thus, we took advantage of this scattering to analyze the light scattered by the spheroids in the forward direction to determine the mean velocities inside the aggregate. The technique is simple, but its explanation would considerably complexify the article. For this reason we refer to the published articles in the manuscript (Brunel, 2017 and Brunel, 2020). Nevertheless, we agree that it is important to quickly explain the technique and to show an example of raw data. Thus, we have added a full subsection "Dynamic Light Scattering and Motile Activity inside MCS" to the Appendix of the revised version.

Figure 5: statistical analysis should be included for D and F.

Statistical analysis has been added to the revised version

Figures 4 and 5: multiple levels of solid stress would be ideal to show the trend.

The choice of using a pressure of 5 kPa was motivated by our previously published experiments (Montel et al., 2012; Brunel et al., 2020) and reported in Author response image 1. Such experiments showed that 5 kPa is the value at which the effect of pressure on proliferation saturates. At lower pressure proliferation and motility inhibition is less clear. Larger pressures do not exacerbate the effect, but appsssssroach the regime of high osmotic pressures, where other cell responses are involved.

We modified the text to better explain these two points. Statistical tests have been added to all plots. We did few experiments at different pressures, which showed qualitatively similar results. However, a precise quantification of the influence of the pressure magnitude in all our experiments would require large statistics and a quantity of work that is beyond our capabilities in a decent amount of time.

Figure 5: do cells extend beyond the periphery of the Matrigel beads?

Cells at the interface often extend outside the MG. Those cells have been excluded from the analysis. Cells shown in the figures do not extend beyond the periphery of MG beads. A sentence was added to the revised manuscript, to precise this point.

3) Mechanistic insights – there are really no mechanistic insights provided as to how increased solid stress might lead to some of the behaviors observed. This is unusual for eLife papers. Even a screen of implicated small molecule inhibitors (e.g. actin, myosin, Rho,.…) could be valuable in generating some general idea into mechanism.

As suggested by the reviewer, we have realized compression experiments after exposure to drugs affecting the cytoskeleton organization and the contractility: Y-27632, Blebbistatin, Cytochalasin D, Nocodazole and Taxol. Unfortunately, cells have no long-term viability, when exposed to those drugs so it was not possible to investigate their impact on cell proliferation and motility. However, it was possible to evaluate how those perturbations affect the compressibility of spheroids to an osmotic compression.

In the experiments, we first expose the MCS to the drug and measure the volume change. Then, we add the big dextran at 5 kPa and measure again the evolution of the MCS volume. We obtained two results:

1) Drugs modify the MCS volume in different manners (see Appendix 1—figure 5) as compared to the control (DMSO). This is in agreement with the fact such pharmacological perturbations are known to impact the single cell volume in different manners (see [Stewart et al., 2011]). Although several mechanisms linking the cytoskeleton and the cell membrane effective permeability have been discovered [Cadart et al., 2019], a theoretical model that would unify these observations is however still lacking to our knowledge. In addition, these drugs can impact not only cell volume, but the compaction of the spheroid by means of cell-cell interactions and contractility thus modify the overall spheroid volume.

2) When dextran is added to the solution, the spheroids get compressed from their initial state (with the drug); such final compression (Dextran+Drug) is comparable to that obtained with dextran alone but the amount of compression with respect to the initial state varies depending on the drug.

Those results are compatible with our idea that the 5 kPa Dextran compression reduces almost to the maximum the inter-cellular space and that cells are then almost fully connective in the final state. Thus, depending on the amount of compression that the drug first creates, the ensuing compression with Dextran will change depending on the available inter-cellular space that remains. For instance, in the presence of cytochalasin, the extra-cellular space is already largely reduced compared to DMSO so when the osmotic compression follows, there is hardly no intercellular space which can still be compressed. This seems to be an additional indication that the MCS compressibility in response to a gentle osmotic pressure is more related to the rheology of the extracellular space than to the internal organization and contractility of the cytoskeleton as we argue in [Dolega et al., 2020]. However, it is difficult to extract from this data a potential mechanism that would explain the effect of mechanical stress in the inter-cellular ECM on the cells motility and proliferation.

Some preliminary experiments indicate that membrane tension evolve under pressure (see also https://www.biorxiv.org/content/10.1101/2021.01.22.427801v1) and may play a major role. Although promising, we still do not have sufficient evidence to support this hypothesis unequivocally.

We don't think these new data add much value to the article, but instead may complicate the message. Thus, unless the reviewer has a different opinion, we added them as additional information.